# Growth anisotropy of the extracellular matrix shapes a developing organ

Stefan Harmansa [1,6], Alexander Erlich[1,2,5,6], Christophe Eloy[2], Giuseppe Zurlo[3] & Thomas Lecuit [1,4] ✉

Final organ size and shape result from volume expansion by growth and shape changes by contractility. Complex morphologies can also arise from differences in growth rate between tissues. We address here how differential growth guides the morphogenesis of the growing *Drosophila* wing imaginal disc. We report that 3D morphology results from elastic deformation due to differential growth anisotropy between the epithelial cell layer and its enveloping extracellular matrix (ECM). While the tissue layer grows in plane, growth of the bottom ECM occurs in 3D and is reduced in magnitude, thereby causing geometric frustration and tissue bending. The elasticity, growth anisotropy and morphogenesis of the organ are fully captured by a mechanical bilayer model. Moreover, differential expression of the Matrix metalloproteinase MMP2 controls growth anisotropy of the ECM envelope. This study shows that the ECM is a controllable mechanical constraint whose intrinsic growth anisotropy directs tissue morphogenesis in a developing organ.

During animal development, tissues are shaped by mechanical forces[1,2]. While active contractile forces have gained substantial attention, the role of cellular volume growth in shaping epithelial tissues remains less explored[3,4]. Epithelial tissues drastically grow in size due to cell growth (cellular increase in mass and volume) and proliferation (increase in cell number)[5], both processes contributing to organ growth. The growth rate measures an increase in tissue mass and volume over a time interval. Tissue growth rates may differ between regions of tissue or between tissues, a situation referred to as differential growth. Differential growth leads to a geometric incompatibility[6–8] which is the source of residual stress, i.e., the stress that remains in the absence of external forces.

Mechanical stresses that arise due to differential growth directly affect and guide the shaping of growing structures. Examples of 3D shape changes driven by differential growth in the plant world are the formation of flower petals[9] and leaves[10]. In the animal world, differential growth can be seen, for example, in the formation of solid tumours[11]. A special case of differential growth is seen in multi-layered tissues such as in organs, where differential growth between adjacent layers can lead to mechanical stress that shapes tissues in 3D. For example, differential growth between connected cell layers drives the folding of the human cortex[12,13], the looping of the gut and formation of villi in the chick[14–16], morphogenesis of the airways[17] and the heart[18]. In a more general context, ideas from differential growth-driven morphogenesis have inspired research into additive manufacturing in an effort to programme arbitrary shapes based on the differential growth of elastic bilayers[19,20].

Here, we use the *Drosophila* wing imaginal disc, a multi-layered epithelial structure to study how the growth and elastic deformation of individual layers affect the typical domed shape of the wing disc. The disc is composed of two stacked epithelial mono-layers: the bottom pseudostratified disc proper epithelium (DP) and the overlying squamous peripodial epithelium (PPE, see Fig. 1a). The basal side of each monolayer of this epithelial 'sandwich' is surrounded by an extracellular matrix (ECM), effectively making the wing disc a four-layer structure. In particular, the ECM has recently gained more attention and was shown to be required for controlling growth[21] and morphology of the disc[22,23].

[1]Aix-Marseille Université & CNRS, IBDM—UMR 7288 & Turing Centre for Living Systems (CENTURI), Campus de Luminy case 907, 13288 Marseille, France. [2]Aix-Marseille Université, CNRS, Centrale Marseille, IRPHE, Turing Centre for Living Systems, Marseille, France. [3]School of Mathematical and Statistical Sciences, University of Galway, University Road, Galway, Ireland. [4]Collège de France, 11 Place Marcelin Berthelot, Paris, France. [5]Present address: Université Grenoble Alpes, CNRS, LIPHY, 38000 Grenoble, France. [6]These authors contributed equally: Stefan Harmansa, Alexander Erlich. ✉e-mail: thomas.lecuit@univ-amu.fr

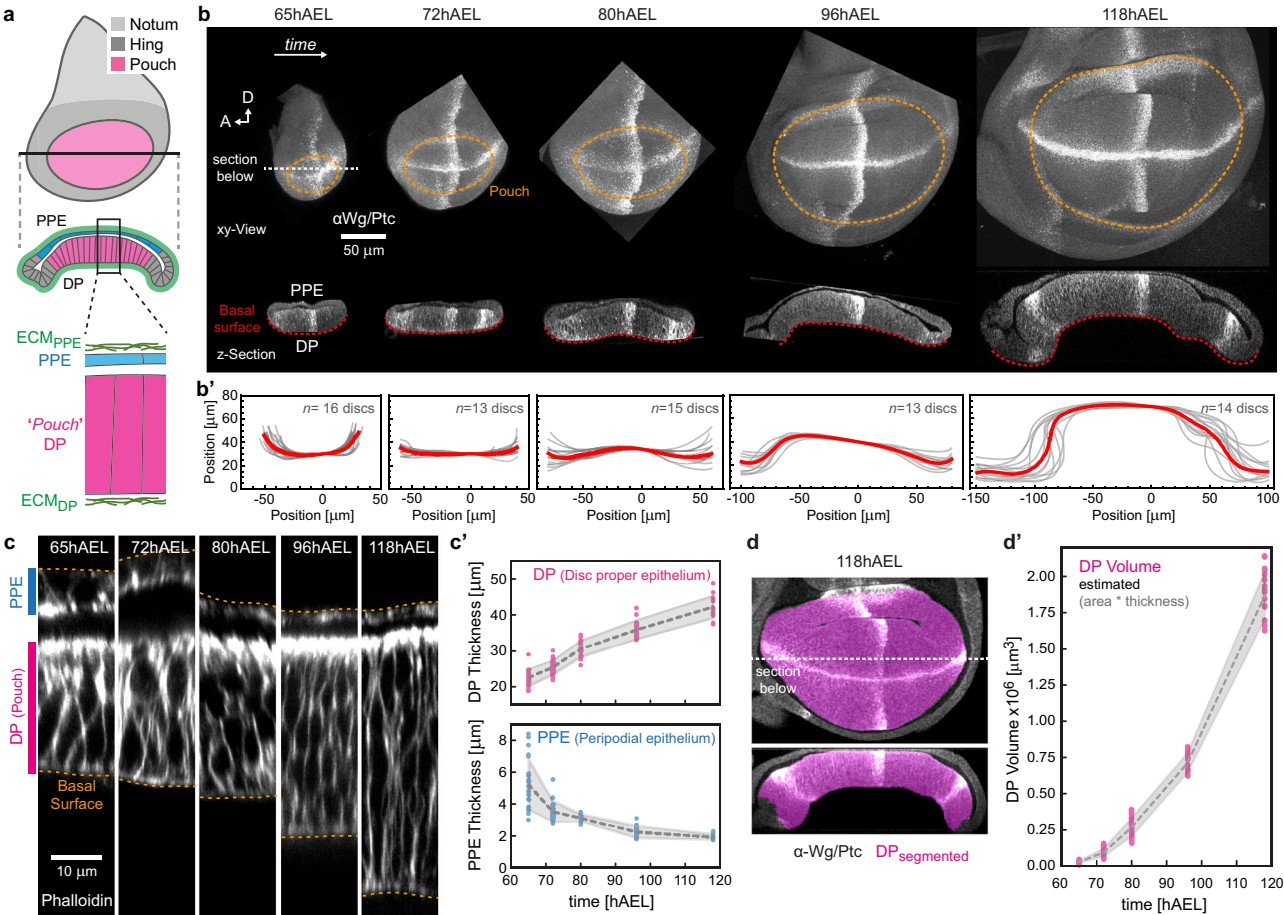

**Fig. 1 | Growth-associated epithelial doming and thickening. a** Scheme of the *Drosophila* wing disc: The disc proper epithelium (DP) is overlaid by the peripodial epithelium (PPE, blue). The DP contains the wing pouch (magenta), surrounded by the hinge (dark grey). The disc is covered by an ECM shell (green). The scheme is adapted from ref. [79], licensed under CC BY 4.0. **b** Wing discs of indicated age (given in hours after egg laying, hAEL) in-plane (top) and section view (bottom) stained for Wingless and Patched (Wg/Ptc) to provide landmarks and the tissue outline. Discs in-plane view are oriented with anterior (A) facing left and dorsal (D) facing up, the pouch is outlined by a dashed line. In section view the PPE is facing upwards. The basal surface of the DP is marked (red dotted line). **b'** Quantification of average basal surface shape (as indicated by the dotted red line in (**b**)). Individual outlines are shown in grey, the average outline in red. **c** Section views of wing discs (at the A/P–D/V boundary intersection) at indicated age classes. Cell outlines are marked by Phalloidin (F-Actin). **c'** Quantification of PPE and DP thickness (measured in sections as shown in (**c**)). Error band indicates standard deviation. **d** A 118hAEL wing disc marked for Wg/Ptc (grey) after segmentation of the DP pouch volume (magenta) in-plane (top) and cross-section view (bottom). **d'** Quantification of pouch volume (see Supplementary Fig. 1 and 'Methods' for details, error bands indicate standard deviation, $n_{65} = 16$, $n_{72} = 13$, $n_{80} = 15$, $n_{96} = 13$, $n_{118} = 14$). Source data are provided as a Source Data file.

ECMs are required for cell polarisation and lumen formation[24,25]. They play an essential role in branching morphogenesis[26,27], provide a mechanical and geometric scaffold for the formation of organoids[28] (reviewed in ref. [29]) and embryonic structures in stem cell-derived systems[30]. More importantly, a growing epithelial tissue constrained by a passive ECM layer was shown to drive tissue bending during optic cup morphogenesis in the chick eye[31] and was speculated as a mechanism for tissue bending in echinoderm gastrulation[32], mammalian lung branching[27,33] and salivary gland morphogenesis[34,35]. These studies suggest a universal mechanism where a passive ECM layer constrains epithelial growth and controls organ morphogenesis.

Here, we demonstrate that during *Drosophila* wing disc doming the ECM acts not as a passive constraint, but as a controllable, actively growing dynamical constraint to the tissue layers. Differential growth between the tissue layers and the bottom ECM layer, with the tissue layers growing more than the ECM layer, results in the accumulation of residual stress leading to elastic deformation and bending of the disc. This is comparable to the functioning of bimetal thermostats, where the differential thermal expansion properties of two connected metal layers lead to bending. We further show that differential growth anisotropy (i.e., anisotropic expansion in 3D) in the epithelium versus the

ECM, controlled by the matrix metalloproteinase MMP2 in the ECM, further fine-tunes the 3D shape of the disc. Hence, control over the degree of growth anisotropy in the ECM provides an elegant and possible universal mechanism for budding, invagination and folding processes during animal development.

## Results

### The disc bends during development

In order to study the 3D morphology of the growing wing disc we focused on the major growth phase from 65 h after egg laying (hAEL, mid-second instar) to the end of larval development (corresponding to 120hAEL at 25 °C). We assessed 3D morphology in cross-sections parallel to the anterior/posterior (A/P) axis. While at 65hAEL the basal surface of the DP curves upwards, it becomes flat around 72hAEL and then starts to form an inverse, dome-like structure after 80hAEL (Fig. 1b bottom, b'). Coinciding with tissue doming, the thickness of the DP epithelium doubles between 65 and 118hAEL. In contrast, the thickness of the PPE decreases from ~5 to ~2 μm during the same period of development (Fig. 1c). Associated with these morphological changes we observed a striking increase in tissue volume. We measured the temporal changes in volume of the central portion of the DP, the wing

pouch, and found a ~66-fold volume increase between 65 and 118hAEL (see Fig. 1d, Supplementary Fig. 1a–g and 'Methods' for details).

Consistent with the tissue scale thickening of the DP epithelium we observed that DP cell width decreases while cell height increases along the apical–basal axis. The inverse is observed for peripodial cells, which decrease their cell height and flatten towards the end of larval development (Supplementary Fig. 2).

Together, our data show a concomitant increase in tissue size with the thickening and bending of the disc's proper epithelium. This raises the question of how growth and tissue morphology are linked during wing disc morphogenesis.

## Tissue thickening is not due to differential cell growth in the tissue plane

We first investigate what drives the doubling in DP thickness. Tissue thickening of the DP could theoretically result from two scenarios: either cells actively increase their height via z-growth or the observed increase in cell height is a result of elastic compression due to cell crowding. Previous work proposed that cell proliferation is increased in the centre versus the periphery of the disc, leading to crowding and stress accumulation in the disc centre[36]. Consistently, artificially increasing growth in clones of cells leads to the accumulation of stress in neighbouring cells though not to an increase in thickness[37,38]. We therefore first considered that the doubling in tissue thickness stems from cellular compression due to spatially non-uniform growth in the plane of the DP tissue.

In order to test if growth is inhomogeneous in the wing disc, we assessed cell growth experimentally using 3 independent methods. We first investigated the spatial pattern of cell divisions by clonal assays (as in ref. [36]), making sure that clonal density is low (~9 clones/disc at 72hAEL, see 'Methods') and hence fusion unlikely to occur. We found no evidence of differential growth in the plane of the disc epithelium from 48-96hAEL (Supplementary Fig. 3a). Also, staining for Phospho-Histone H3, a marker for mitosis (Supplementary Fig. 3b), yielded a uniform proliferation density from 72 to 80hAEL, and even a decreased central proliferation density at 96hAEL.

Importantly, growth is a dynamic and 3-dimensional process. We therefore established volumetric 3D live-imaging in ex vivo cultured wing discs to directly measure clonal growth rates (see 'Methods' and Supplementary Fig. 3c–f). We have followed volumetric changes in small clones containing few cells for up to 10 h in disc explants at 72hAEL, the time point when DP thickness starts to increase (Supplementary Fig. 3c). Consistently, the cell growth rate was uniform (0.085 h$^{-1}$) within the DP epithelial plane (Supplementary Fig. 3d, e).

In conclusion, our results indicate that growth is fairly uniform in the plane of the DP epithelium at different stages of larval development. Therefore, tissue thickening has a different origin than differential cell growth in the plane of the DP epithelium.

## Bending is not due to differential growth of the tissue layers

Next, we considered the 3D multi-layered structure of the disc consisting of two growing tissue layers: the DP and the PPE. We established a non-linear elastic continuum model[39,40] based on the concept of Morphoelasticity[41–43] and simulated two mechanically coupled and growing elastic layers (DP and PPE, see Fig. 2a) using the finite element method[39,40]. Morphoelasticity considers that changes in tissue shape (the total deformation), described by the deformation gradient tensor $\mathbb{F}$, result from both an increase in volume (i.e. growth) and elastic deformations. Indeed, $\mathbb{F}$ captures the transformation of a growing structure from its original stress-free state (early in development), called the initial state, to its grown, final shape and size (observed state, Fig. 2a). Importantly, this transformation is due to a growth component, described by the growth deformation tensor $\mathbb{G}$, and an elastic component, described by the elastic deformation gradient tensor $\mathbb{A}$ (such that $\mathbb{F} = \mathbb{A}\mathbb{G}$).

We assumed that in the initial state, the two tissue layers are stress-free (Fig. 2a). The growth tensors $\mathbb{G}_{PPE}$ and $\mathbb{G}_{DP}$ describe the growth of the PPE and the DP layers, respectively. When growth between the two layers differs, namely $\mathbb{G}_{PPE} \neq \mathbb{G}_{DP}$, the two grown tissue layers have different sizes (geometric incompatibility, see Fig. 2a reference state). However, the two different sized discs can be connected into one coherent object by elastic deformation (by $\mathbb{A}$) which builds residual stress. The assumption that the wing disc is residually stressed is supported by the fact that cutting the domed wing disc at 3rd larval stage lead to nearly instantaneous relaxation of the shape (Supplementary Fig. 4a). We described the material of the layers by a nearly incompressible neo-Hookean material, a commonly used material model for morphogenetic tissue due to its relative simplicity and ability to describe large deformations[39,40,44]. In a polar cylindrical basis $\{E_R, E_\theta, E_Z\}$, we denote components of the growth tensor $\mathbb{G}$ by $[\mathbb{G}]$ in the DP and PPE as

$$[\mathbb{G}_{DP}] = \mathrm{diag}(\gamma_{DP}, \gamma_{DP}, \gamma_{Z,DP}), \quad [\mathbb{G}_{PPE}] = \mathrm{diag}(\gamma_{PPE}, \gamma_{PPE}, \gamma_{Z,PPE}), \quad (1)$$

respectively. Here, $\gamma_{DP}$, $\gamma_{PPE}$ are the in-plane growth parameters in the respective layers (see coordinate system in Fig. 2a). The growth parameters in the axial, or Z-direction, are denoted $\gamma_{Z,DP}$, $\gamma_{Z,PPE}$. When $\gamma_{DP} = \gamma_{Z,DP}$, then growth is isotropic, i.e. the same in all directions. In the case $\gamma_{DP} \neq \gamma_{Z,DP}$, growth is anisotropic. A subcase of anisotropic growth is planar growth, where $\gamma_{Z,DP} = 1$. Notice that we do not consider polar growth anisotropy, in which case the radial and hoop components of the growth tensor, i.e., the first two entries of the diagonal matrices in Eq. (1), would have been different one from the other.

We assumed planar growth for the two tissue layers, since in epithelia cell divisions occur in the epithelial plane[45] ($\gamma_{Z,PPE} = \gamma_{Z,DP} = 1$). When the two layers grow equally ($\gamma_{PPE} = \gamma_{DP}$), growth is compatible and does not induce elastic stress or bending (Fig. 2b, left). In contrast, when the PPE grows faster than the DP layer ($\gamma_{PPE} > \gamma_{DP}$), we found that qualitatively correct bending is accounted for (Fig. 2b, right). In order to obtain bending that is comparable to wing discs at 118hAEL the model required that the PPE layer grew ~70% more in volume than the DP layer (see the overlay, Fig. 2b top and supplementary information Section M1.2.1).

In order to experimentally test this prediction, we measured growth rates in DP and PPE clones in ex vivo cultured discs (Fig. 2c). However, at the time point of bending onset (72hAEL) clonal growth rates over a period of 9 h were comparable in both epithelial layers. We next quantified the temporal volume increase of the PPE tissue that covers the DP pouch region between 72 and 118hAEL (Supplementary Fig. 1h–j). Interestingly, we found that after 72hAEL the PPE volume increases significantly less than the DP volume (Supplementary Fig. 1k), suggesting that the PPE layer possibly acts as a constraint for DP expansion. To test the mechanical role of the PPE layer, we further reduced peripodial growth by overexpression of a dominant-negative form of Phosphatidylinositol 3-kinase (PI3K$_{DN}$), a major growth regulator. However, reduced peripodial growth did not abolish disc bending (Supplementary Fig. 4b–h).

These results demonstrate that epithelial thickening and the direction of doming cannot be explained by non-uniform growth within or between epithelial layers. In particular, our finding that volume growth is reduced in the top (PPE) versus the bottom layer (DP) suggests that epithelial growth dynamics might favour inverse bending of the disc tissue. This raises the question which other component/constraint is responsible for guiding the upward doming of the wing disc.

## The ECM is essential for tissue bending and thickening

Given that the growth dynamics of the tissue layers cannot explain disc morphology, we next considered the ECM layers. The basal surface of the wing disc is covered by a sheet-like ECM shell that is rich in

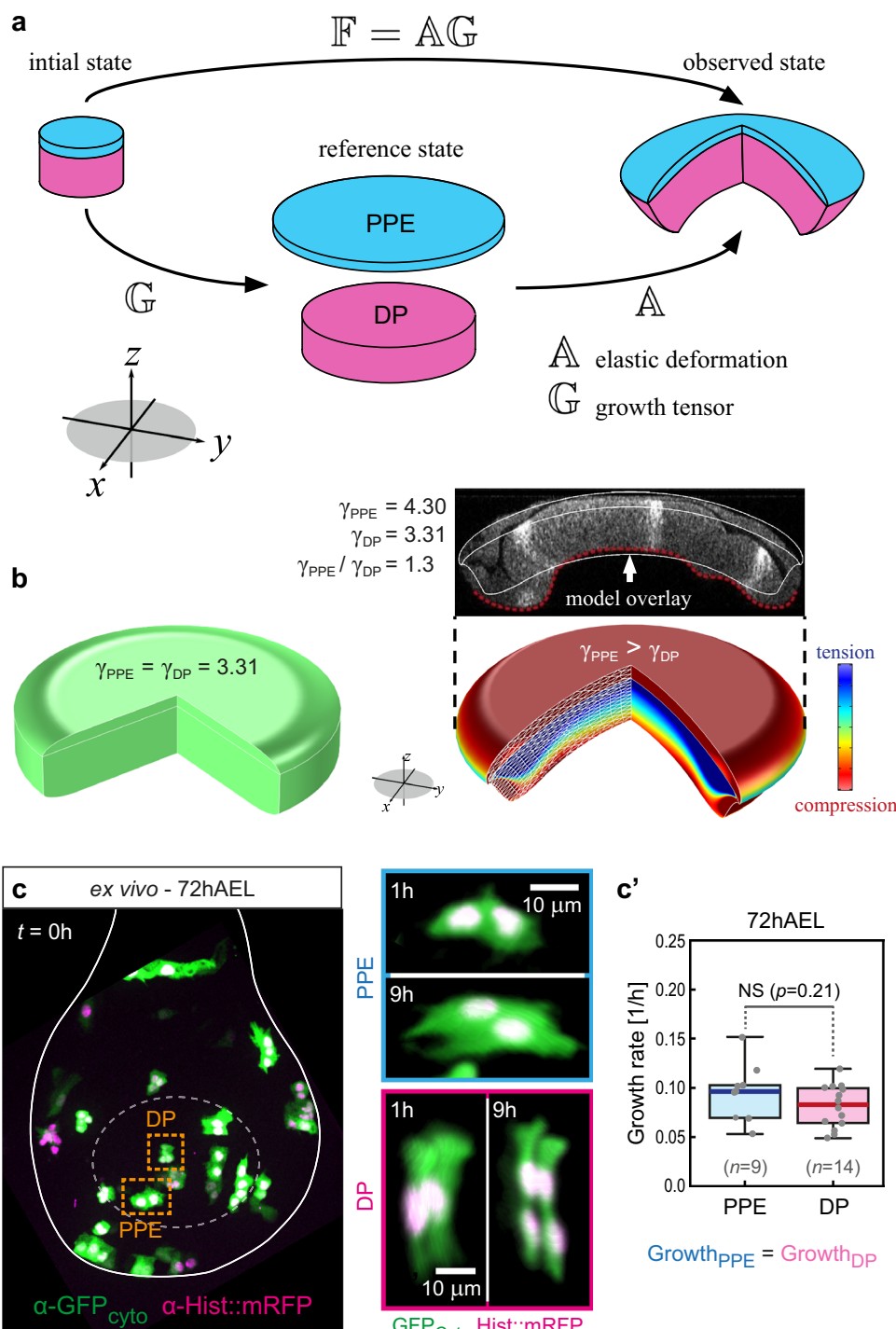

**Fig. 2 | Non-uniform growth between epithelial layers does not drive tissue bending. a** Geometric decomposition of the wing disc as a sandwich of growing elastic layers, the PPE (blue) and the DP (magenta), each growing uniformly in-plane. The growth tensor $\mathbb{G}$ describes growth without stress; different growth rates in the respective layers lead to different sizes of the individual discs. Elastic deformation (described by $\mathbb{A}$) connects the discs into one coherent object, leading to the observed morphology and accumulation of residual stress. **b** Left: if the growth rates in all layers are equal ($\gamma_{DP} = \gamma_{ECM}$ in Eq. (1)), growth is compatible, leading to no residual stress (green colour). Right: if the PPE volume grows ~70% times faster than the DP, simulations approximate well the curvature and morphology of 118hAEL wing discs (see inset with model overlay), with the PPE being in compression (red). **c** Left: maximum projection of an ex vivo cultured 72hAEL wing disc at the start of imaging (0 min) containing clones of cytosolic GFP (volume) and Histone::RFP. Right: 3D projection of two example clones at the beginning and end of culture. **c'** Average clonal growth rates at 72hAEL. In the box plots, the median is indicated by a central thick line, while the interquartile range (containing 50% of the data points) is outlined by a box. Whiskers indicate the minimum and maximum data range ($n_{PPE}$ = 9 clones/4 discs, $n_{DP}$ = 14 clones/ 3 discs). Statistical significance was assessed by a two-sided Student's t test (unequal variance). Source data are provided as a Source Data file.

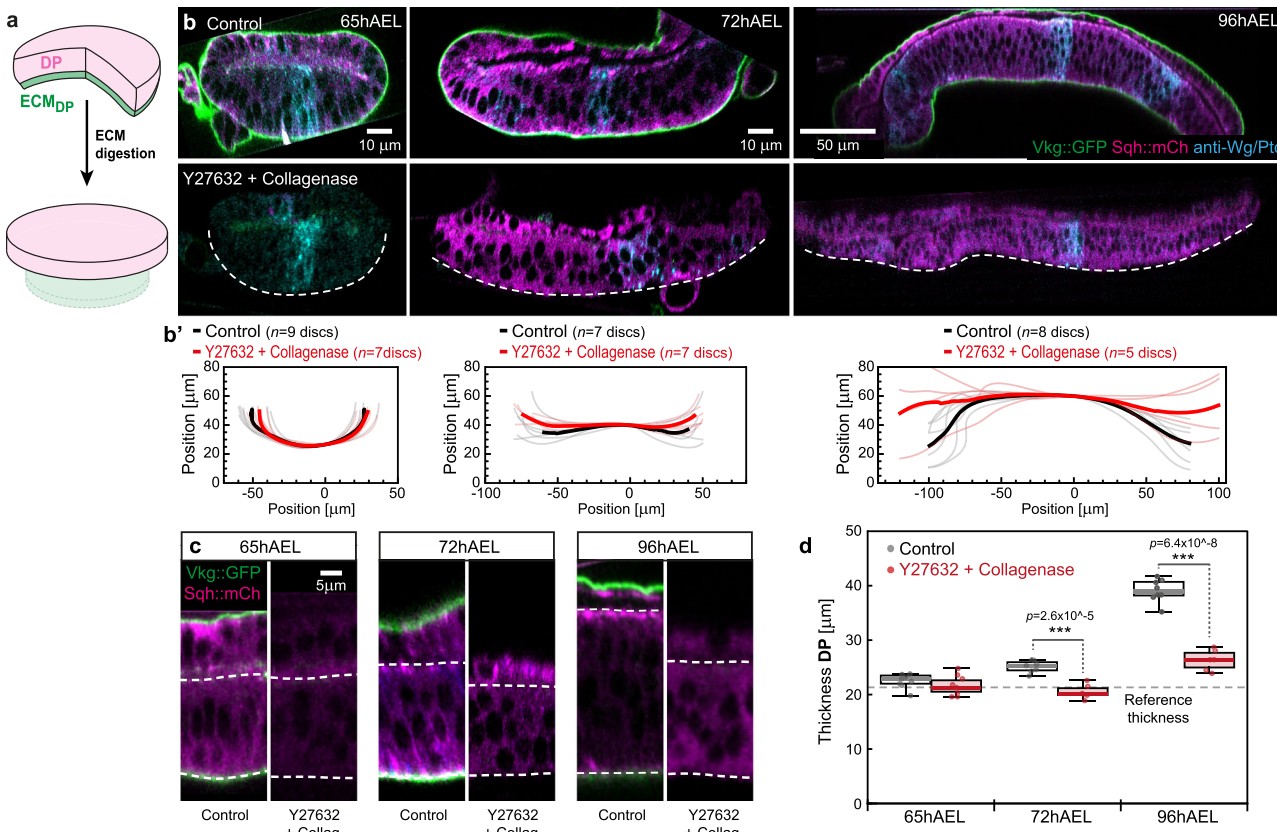

**Fig. 3 | The ECM constraints *planar* epithelial growth and is required for disc doming. a** Acute digestion of the ECM shell by Collagenase leads to relaxation of the epithelial layers. **b** Top: representative sections of control wing discs at indicated stages of development marked for the ECM (Vkg::GFP, green) and MyoII (Sqh::mCh, magenta). Bottom: representative wing discs after acute digestion of the ECM by Collagenase and inhibition of MyoII (by Y-27632, see 'Methods'). Dashed line marks DP basal outline. **b'** Average basal outlines before (black line) and after ECM digestion + Myosin II inhibition (red line). **c** Cross-sections of control and ECM digested + MyoII inhibited wing discs. Apical and basal surface of the DP are marked by dashed lines. **d** Quantification of DP epithelial thickness (as seen in **c**) in control discs (black dots) and upon digestion of the ECM and subsequent inhibition of MyoII (red dots). The reference thickness observed at 65hAEL is indicated by a dashed line (*n*-numbers are indicated in **b'**). In the box plots the median is indicated by a central thick line, while the interquartile range (containing 50% of the data points) is outlined by a box. Whiskers indicate the minimum and maximum data range. Statistical significance was assessed by a two-sided Student's *t* test (unequal variance, ***$P \leq 0.0005$). Source data are provided as a Source Data file.

Collagen IV, called Viking (Vkg) in *Drosophila*[46]. Given that *vkg* mutations[47] or acute loss of the ECM[21,22] impair tissue thickening and doming, we next tested if the ECM constrains tissue growth.

A continuous layer of ECM, visualised by a GFP-tagged version of Vkg (Vkg::GFP)[48] covered the basal surface of the DP and PPE (Fig. 3b, top). In order to investigate the role of this ECM shell, we acutely removed the ECM by Collagenase digestion at different stages of development (Fig. 3a). At 96hAEL, when the disc is fully domed, this led to an inversion of tissue curvature, consistent with earlier reports[22] (Supplementary Fig. 5a, b). We reasoned that residual stress due to either reduced PPE growth or apical Myosin II (MyoII) contractility would underlie this inverted curvature. Acute ECM digestion with concomitant inhibition of MyoII led to a complete flattening of the epithelial layers (Fig. 3b bottom, b'), demonstrating that apical MyoII contractility is responsible for the observed inverted curvature. Furthermore, this result demonstrates that the PPE layer (despite its reduced growth) cannot induce an inverse doming and hence its mechanical impact is negligible for disc morphology (see discussion). Importantly, MyoII inhibition alone did not affect overall disc morphology (Supplementary Fig. 5a, b). At 65hAEL, collagenase treatment left the disc in its flat configuration. These findings show that MyoII is negligible, but that the ECM layer is essential for wing disc bending.

In addition, we also observed a significant decrease of epithelial thickness upon loss of the ECM and MyoII contractility after 72hAEL (Fig. 3c, d and Supplementary Fig. 5c, d). At all investigated time points, digestion of the ECM returned epithelial thickness close to the values observed in control discs at 65hAEL. In contrast, at 65hAEL no thickness relaxation upon ECM digestion was observed.

The fact that the relaxed tissue thickness remains nearly constant over time indicates that the DP layer does not actively increase its thickness by growth along the *z* axis (see also Supplementary Fig. 6). Therefore, our data indicate that the DP layer predominantly grows in-plane and that the observed thickness increase is mainly due to elastic compression mediated by the ECM shell covering the whole disc.

## Differential volume growth between DP and associated ECM

Tissue bending requires an intact ECM layer, however, the ECM is present on both sides of the disc, facing the DP and PPE tissue layers. This suggests that the top and bottom ECM layers must grow differently. We therefore next considered the growth of the two ECM layers. Further, we hypothesised that a volumetric growth mismatch between the DP epithelium and the bottom ECM_DP layer is responsible for the build-up of stress and the observed morphology.

We first investigated the thickness of the ECM, visualised by Vkg::GFP, at defined stages of development. We found that the thickness of the top ECM_PPE layer did not significantly change from 72 to 118hAEL (Supplementary Fig. 7a, b). In contrast, the thickness of the bottom ECM_DP increased by ~36% during this time window (Supplementary Fig. 7c).

Next, we quantified the growth of the bottom $ECM_{DP}$, focusing on the wing pouch (marked by the ring of Wingless expression), following two approaches. First, we estimated the $ECM_{DP}$ volume by multiplying the ECM thickness with the pouch area (Supplementary Fig. 7d, see 'Methods' for details). Secondly, we measured the increase in integrated fluorescence intensity of the Vkg::GFP signal underlying the wing pouch (Supplementary Fig. 8c). Comparing the relative increase in DP volume with the increase in $ECM_{DP}$ estimated volume or integrated intensity showed that the DP layer outgrew the $ECM_{DP}$ by ~15–20% (Supplementary Figs. 7d and 8d). In contrast, when quantifying the changes in integrated Vkg::GFP intensity in the top $ECM_{PPE}$ we observed that the top $ECM_{PPE}$ integrated intensity increases ~2.5-times faster than the PPE tissue volume (Supplementary Fig. 8a, b and see 'Discussion'). Notably, we observed a very similar increase in top $ECM_{PPE}$ integrated intensity and DP volume (Supplementary Fig. 8b), suggesting that the top ECM layer grows compatible with the DP tissue layer.

In summary, these findings show that the DP tissue layer outgrows the bottom $ECM_{DP}$ layer, leading to an effective volumetric growth mismatch. Modelling a 20% growth mismatch was however not sufficient to account for correct tissue geometry (see Supplementary Fig. 8e). In contrast, the top $ECM_{PPE}$ layer shows similar growth dynamics as the DP layer suggesting that the top $ECM_{PPE}$ layer does not act as constraint for disc bending due to its compatible growth. This supports our hypothesis that the bottom $ECM_{DP}$ acts as the major geometric constraint for epithelial growth.

## Spatial differences in ECM growth anisotropy

Since the observed volumetric growth mismatch between DP and $ECM_{DP}$ is not sufficient to explain disc morphology we next considered a difference in growth anisotropy between the $ECM_{DP}$ layer ($\gamma_{Z,ECM} \neq 1$) and the DP tissue layer which grows in-plane ($\gamma_{Z,DP} = 1$). Further, we hypothesised that the top $ECM_{PPE}$ and bottom $ECM_{DP}$ layers have different growth anisotropy.

To assess ECM growth anisotropy we eliminated the contribution of elastic deformation (due to epithelial growth) and visualised the relaxed ECM layers in their unstressed configuration ('reference state'). This was achieved by exposing disc explants to the detergent Triton X-100 (referred to as 'decellularization', see 'Methods') which resulted in lipid bilayer degradation and a loss of cellular hydrostatic pressure acting on the surrounding ECM (Fig. 4a, b).

We first investigated changes in relaxed ECM thickness upon decellularization during development. Strikingly, the relaxed thickness of the top $ECM_{PPE}$ did not change during larval development (Fig. 4c top and Supplementary Fig. 9a), while the relaxed thickness of the bottom $ECM_{DP}$ increased by ~40% between 72 and 118hAEL (Fig. 4c bottom and Supplementary Fig. 9b). Therefore, the top $ECM_{PPE}$ layer follows planar growth ($\gamma_Z = 1$). In contrast, the growth of the bottom $ECM_{DP}$ is markedly non-planar ($\gamma_Z \neq 1$), as it grows in thickness as well as in the plane.

To monitor the elastic relaxation of the ECM, we decellularized disc explants ex vivo and used live imaging to follow changes in ECM area and thickness. Due to a lack of traceable landmarks in the ECM we bleached circular regions of interest (ROI) on the Vkg::GFP-labelled ECM (Fig. 4d, see 'Methods'). Following decellularization, a previously stretched ROI is expected to relax in area and to thicken concomitantly. In the top $ECM_{PPE}$ neither the circular area nor ECM thickness changed once the $ECM_{PPE}$ reached an equilibrium configuration after 60 min (Supplementary Fig. 9d). This confirmed that the upper $ECM_{PPE}$ layer is not under mechanical load and grows compatible with the doming DP layer. A similar analysis in the bottom $ECM_{DP}$ layer showed however that the bleached circular area decreased to ~79% of the original area after one hour (Fig. 4d' and Supplementary Fig. 9c). Concomitantly, the bottom $ECM_{DP}$ thickness increased by ~25% (Fig. 4d"), confirming that the bottom ECM layer is stretched

elastically by the growing DP layer. The area decrease and concomitant thickness increase of the bottom $ECM_{DP}$ showed a conservation of volume upon elastic deformations, which means that the $ECM_{DP}$ is incompressible.

In summary, these results demonstrate that the top $ECM_{PPE}$ layer shows planar growth, well compatible with the DP layer. In contrast, growth of the bottom $ECM_{DP}$ is incompatible with the rest of the disc because it also grows in thickness.

## Differential growth anisotropy recapitulates tissue shape changes

In order to test whether such differences in growth anisotropy can quantitatively account for the observed morphogenesis of the disc, we next modified the non-linear elastic bilayer model. Since the $ECM_{PPE}$ grows compatibly with the DP layer its effect on disc mechanics and shape can be neglected and hence the $ECM_{PPE}$ was excluded from further modelling. We also excluded the PPE tissue layer from the modelling since we have demonstrated that its mechanical effect is negligible. Hence, in the following we considered a structure consisting of the DP layer, following planar growth, and the $ECM_{DP}$ layer, following non-planar growth ($\gamma_Z > 1$, Fig. 5a). In the DP tissue layer, growth is planar (see Fig. 3), hence

$$[\mathbb{G}_{DP}] = \operatorname{diag}(\gamma_{DP}, \gamma_{DP}, 1). \quad (2)$$

Here, $\gamma_{DP}$ was determined through a least-square fit of the experimental volumetric data (see Supplementary Information).

The thickness of the bottom ECM layer increases with time and hence follows a non-planar mode of growth. Since the extent of growth anisotropy is not known we introduced a positive *growth anisotropy parameter $\rho$* for the bottom ECM layer:

$$[\mathbb{G}_{ECM}] = \operatorname{diag}(\gamma_{ECM}, \gamma_{ECM}, \gamma_{ECM}^{\rho}). \quad (3)$$

If $\rho = 0$, the ECM would grow in the plane like the DP. If $\rho = 1$, the ECM would grow isotropically. If $\rho > 1$, growth would be primarily in $Z$-direction. We obtained $\gamma_{ECM}$ via a linear fit of the experimental volume data, but $\gamma_{ECM}$ remained a function of $\rho$. Our fitting procedure respects that $\rho$ determines how much volume is deposited in Z vs planar direction without changing the ECM volume (see Supplementary Information Section M1.2.2).

Therefore, two dimensionless parameters required to parameterise the model remained to be determined: the growth anisotropy parameter $\rho$ and the ratio of elastic moduli, $\mu = \mu_{ECM}/\mu_{DP}$, where $\mu_{ECM}$ and $\mu_{DP}$ are the shear moduli of the respective layers. These parameters $\rho$ and $\mu$ were determined by comparing model predictions to three experimentally measured quantities: The DP thickness $H_{DP}$, the ECM thickness $H_{ECM}$, and the relative thickness increase of the $ECM_{DP}$ upon decellularization. Using a phase diagram we determined $\mu = 25$ and $\rho = 0.45$ (grey region in Fig. 5b) as best fit where all three measurables are within tolerance to experimental data (see SI). The latter result suggested that bottom $ECM_{DP}$ growth anisotropy lies midway between planar ($\rho = 0$) and isotropic growth ($\rho = 1$). Figure 5c shows the simulated versus experimentally measured values for this best fit. The thickness values $H_{DP}$, $H_{ECM}$ are given alongside a tolerance interval that quantifies the relative error between experimental means and simulated values. The simulations captured well the trends of the increasing thicknesses. Also the shape of the simulated wing disc (Fig. 5d) fitted well to experimentally obtained morphologies and revealed an increasing build-up of tension in the ECM and compression in the DP.

In summary, a simple mechanical model of growth, incorporating distinct growth anisotropies in the DP and bottom ECM layers, recapitulates wing disc morphogenesis in 3D.

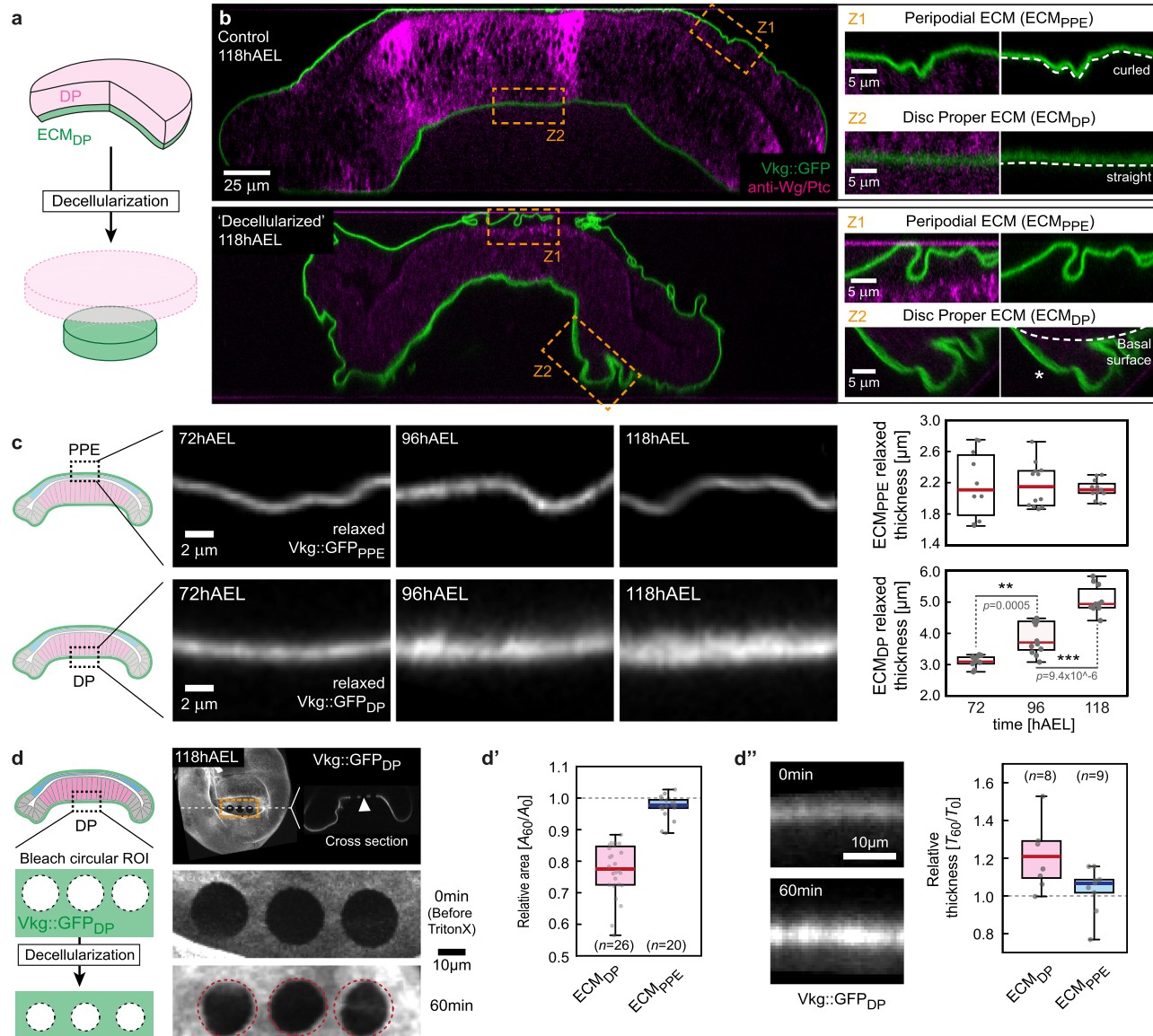

**Fig. 4 | Spatial differences in ECM growth anisotropy amplify stress accumulation and allow symmetry breaking. a** Acute removal of cellular pressure by decellularization reveals the relaxed configuration of the ECM. **b** Cross-sections of a control wing disc (top) and a wing disc after decellularization (bottom, see 'Methods'). The ECM is marked by *Vkg::GFP* (green) and the epithelial layers by a staining for Wg/Ptc (magenta). Right: magnifications of indicated regions. The relaxed geometry of the ECM was assessed in regions where the ECM had separated from the epithelial layer (see asterisk). **c** Cross-sections of relaxed sections of the top $ECM_{PPE}$ (marked by Vkg::$GFP_{PPE}$, top) and the bottom $ECM_{DP}$ (Vkg::$GFP_{DP}$, bottom) at indicated stages. Right: quantification of relaxed ECM thickness (see 'Methods' for details, *n* indicates number of discs, for $ECM_{PPE}$: $n_{72} = 10$, $n_{96} = 12$, $n_{118} = 11$, for $ECM_{DP}$: $n_{72} = 9$, $n_{96} = 12$, $n_{118} = 10$). Statistical significance was assessed by a two-sided Student's *t* test (unequal variance, *$P \leq 0.05$, **$P \leq 0.005$, ***$P \leq 0.0005$). **d** Left: scheme showing how circular regions of interest (ROI) were

bleached on the ECM Vkg::GFP signal. Right: plane and section view of a 118hAEL Vkg::GFP wing disc with 3 bleached circles in the bottom $ECM_{DP}$ (arrowhead). Below magnifications of the circles before and 60 min after decellularization are shown. **d'** Upon decellularization the relative changes in circular area $A$ before ($A_0$) and 60 min after decellularization ($A_{60}$) were quantified in the top $ECM_{DP}$ (magenta) and the bottom $ECM_{PPE}$ (blue, $n$-$ECM_{DP}$ = 26 circles/9 discs, $n$-$ECM_{PPE}$ = 20 circles/10 discs). **d''** Left: cross-section of the bottom $ECM_{DP}$ marked by Vkg::GFP before and 60 min after decellularization. Right: quantification of relative changes in ECM thickness in the bottom $ECM_{DP}$ (blue) and the top $ECM_{PPE}$ (blue) 60 min after decellularization ($n$-$ECM_{DP}$ = 8 discs, $n$-$ECM_{PPE}$ = 9 discs). Statistics: in box plots, the median is indicated by a central thick line, while the interquartile range (containing 50% of the data points) is outlined by a box. Whiskers indicate the minimum and maximum data range. Source data are provided as a Source Data file. The schemes in panels c and d are adapted from ref. [79], licensed under CC BY 4.0.

## MMP2 is required for planar ECM growth

ECM growth anisotropy differs in the PPE and DP layers, raising the question of what controls planar versus more isotropic ECM growth, respectively. We hypothesised that the difference in growth anisotropy is due to differential expression of ECM modifiers in the PPE versus the DP layer. A potential ECM modifier is the matrix metalloprotease 2 (MMP2) which is expressed in the peripodial layer of the

wing[49] and eye disc[50]. MMPs are known for their role in matrix degradation[23,49,51].

We stained wing discs for MMP2 and confirmed that MMP2 levels are higher in the PPE layer compared to the DP layer throughout disc growth (Fig. 6a and Supplementary Fig. 10a). We therefore hypothesised that peripodial MMP2 is required for planar growth of the top $ECM_{PPE}$.

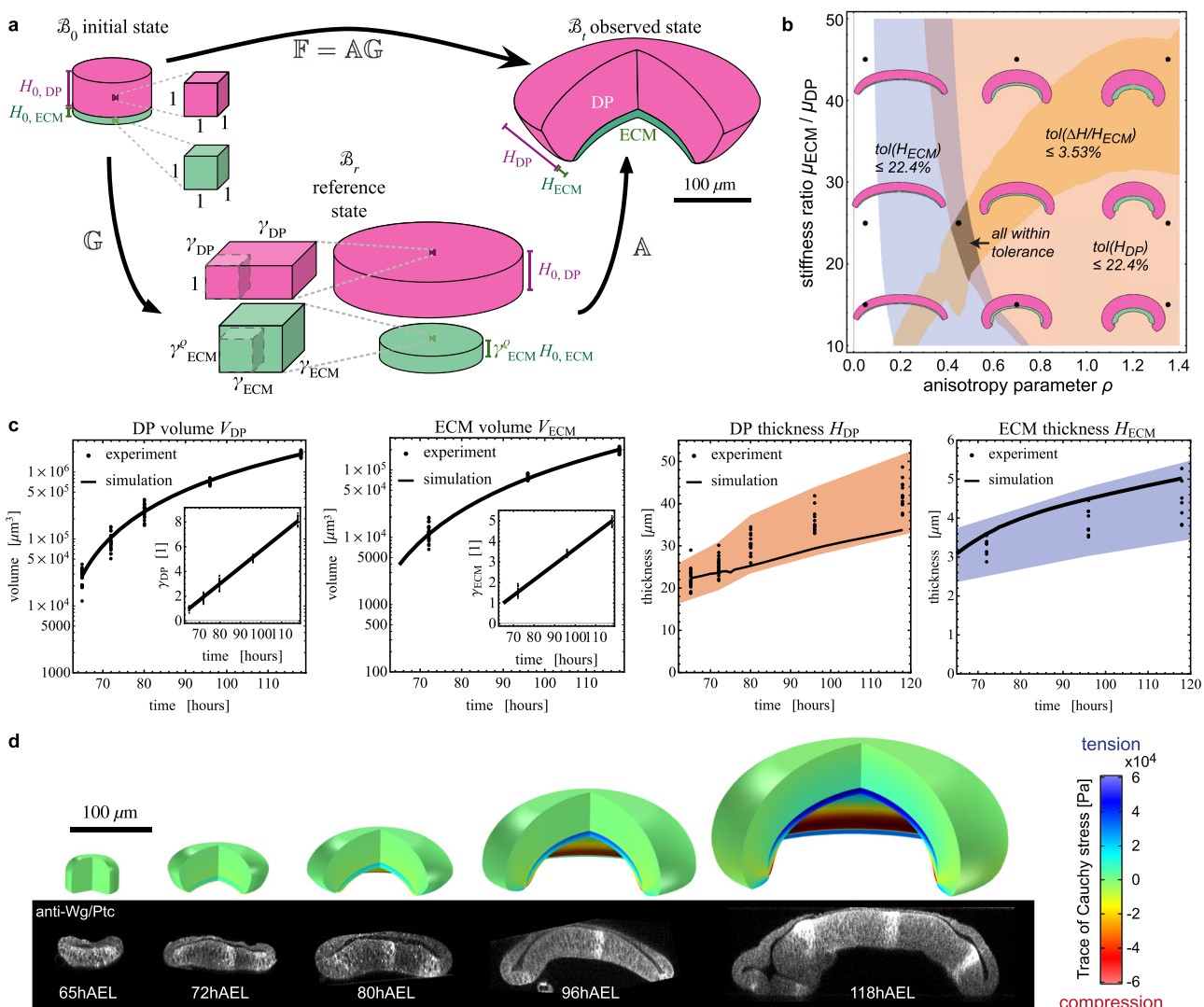

**Fig. 5 | An elastic bilayer model captures growth-induced epithelial morphogenesis. a** Geometric decomposition for a bilayer structure, composed of DP (purple) and ECM$_{DP}$ (green). In the initial state $\mathcal{B}_0$, the disc is unstressed and undeformed. Representative volume elements with unit lateral lengths are shown. The DP and ECM thickness is denoted $H_{0,DP}$ and $H_{0,ECM}$, respectively. The growth tensor $\mathbb{G}$ describes the transformation to the reference state $\mathcal{B}_r$ which is grown and relaxed. Since the DP grows in-plane, its thickness remains $H_{0,DP}$. The ECM grows orthogonally to the plane as controlled by the anisotropy parameter $\rho$, and the relaxed thickness is greater than $H_{0,ECM}$. Through the elastic deformation gradient $\mathbb{A}$, stress is introduced, leading to the domed observed state $\mathcal{B}_t$. **b** Parameter diagram to determine a region in which simulation values are within chosen tolerances of experimental measurements. In the blue, red and orange regions, the simulation results are within tolerance of the measured ECM thickness $H_{ECM}$, DP thickness $H_{DP}$, and relative increase in ECM thickness upon decellularization (see Supplementary Information for details). In the dark region, all three conditions are satisfied simultaneously. Parameters $\rho = 0.45$ and $\mu = 25$ were chosen as the best fit, of which the morphology is shown in the central out of 9 insets showing cross-sections of the simulated wing discs. The result $\rho = 0.45$ suggests that the bottom ECM$_{DP}$ growth anisotropy lies midway between planar ($\rho = 0$) and isotropic growth ($\rho = 1$). **c** Model results ($\rho = 0.45$, $\mu = 25$) compared to experimental data. Left two plots: the insets show linear profiles of $\gamma_{DP}$ and $\gamma_{ECM}$. Right two plots: comparison between simulated and experimental values for DP thickness $H_{DP}$, and ECM thickness $H_{ECM}$. The coloured bands show the region within 22.4% of the mean of experimental values, in which the simulation values are contained. **d** The shape of the simulated wing disc, using the best fit parameters, is shown compared to cross-sections of representative wing discs. The simulated discs are shown with a quarter of each disc removed for illustration purposes (simulations are axisymmetric).

In order to investigate MMP2 function we knockdown (KD) MMP2 in the posterior compartment of the wing disc (referred to as MMP2$_{KD}$, see Supplementary Fig. 10b, c). MMP2$_{KD}$ resulted in a posterior shift of the peripodial anterior–posterior compartment boundary (A/P$_{PPE}$) relative to the A/P boundary in the DP layer (A/P$_{DP}$, see arrowheads in Fig. 6b, c and Fig. 6d, left). This shift coincided with increased epithelial thickness in posterior PPE cells where MMP2 was knocked-down (Fig. 6d, right and Supplementary Fig. 10g–i)) and reduced apical cell surface area (Fig. 6e). These results indicate that MMP2 is required for planar expansion and flattening of PPE cells.

Next, we asked if reduced PPE expansion in MMP2$_{KD}$ discs was due to changes in ECM growth anisotropy. As previously, we used

decellularization to compare the relaxed ECM configuration in MMP2$_{KD}$ and control discs. We decellularized late 3rd instar live explants (ex vivo) to directly follow changes in shape and intensity of the Vkg::GFP-labelled ECM layer, focusing on the ECM$_{PPE}$ posterior to the peripodial A/P$_{PPE}$ boundary (orange bracket in Fig. 6f). Consistent with previous findings, decellularization of control discs did not change ECM$_{PPE}$ thickness (Fig. 6g, i) or Vkg::GFP$_{PPE}$ intensity (Supplementary Fig. 10f). In contrast, in MMP2$_{KD}$ discs decellularization resulted in a significant increase in ECM$_{PPE}$ thickness (Fig. 6h, i) and Vkg::GFP$_{PPE}$ intensity (Supplementary Fig. 10d–f). Furthermore, the relaxed ECM thickness was significantly increased in MMP2$_{KD}$ compared to controls (Fig. 6i). Therefore, a loss of MMP2 modifies

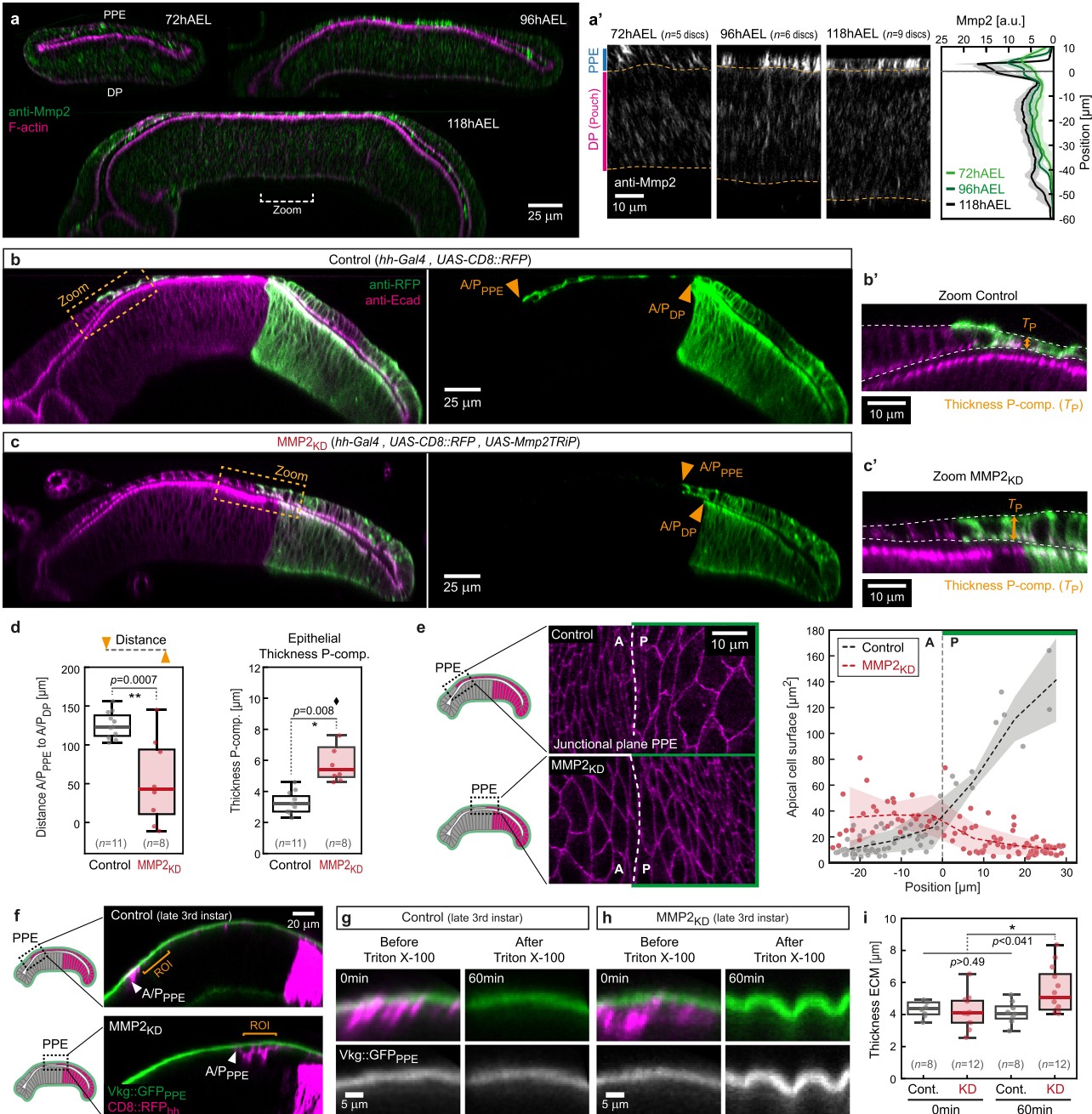

**Fig. 6 | MMP2 modulates peripodial ECM growth anisotropy. a** Section views of wing discs stained for MMP2 (green) at indicated time points and magnifications of the pouch region (**a'**, dashed lines mark DP). Right: quantification of average MMP2 fluorescence intensity distribution (dashed line indicates DP apical surface, error bands indicate the standard deviation). **b** Section view of control wing discs expressing RFP (green) in the posterior compartment. A/P-compartment boundaries are indicated in the PPE (A/P$_{PPE}$) and in the DP (A/P$_{DP}$) by arrowheads. **b'** Magnification of the A/P boundary region, indicated in (**b**). **c** In MMP2$_{KD}$ wing disc the P-compartment in which MMP2 in knocked-down is marked by RFP (green). The A/P boundary region is magnified in (**c'**). **d** Left: quantification of the distance between the A/P boundary in the PPE versus the DP layer. *n* indicates the number of discs. Right: quantification of epithelial Thickness ($T_P$) as indicated in (**b'+c'**). **e** Left: representative magnifications of the peripodial A/P boundary region in *x/y*-view (junctional plane) in control (top) and MMP2$_{KD}$ (bottom). Right: quantification of apical cell surface in the images shown left (error band indicates standard deviation). **f** Section of representative control (top) and MMP2$_{KD}$ (bottom) wing discs expressing Vkg::GFP (green). The ECM$_{PPE}$ was analysed posterior of the peripodial A/P boundary (orange bracket, ROI). **g** Representative posterior section of a control peripodial ECM before (left, 0 min) and after (right, 60 min) decellularization. **h** Section of a MMP2$_{KD}$ ECM before (left) and after (right) decellularization. **i** Quantification of peripodial ECM thickness of indicated conditions (assessed at ROI, *n* indicates number of discs). Statistics: Statistical significance was assessed by a two-sided Student's *t* test (unequal variance, *$P \leq 0.05$, **$P \leq 0.005$, ***$P \leq 0.0005$). In box plots the median is indicated by a central thick line, while the interquartile range (containing 50% of the data points) is outlined by a box. Whiskers indicate the minimum and maximum data range. Source data are provided as a Source Data file. The schemes in panels **e** and **f** are adapted from ref. [79], licensed under CC BY 4.0.

peripodial ECM$_{PPE}$ growth anisotropy from planar to non-planar, 3D growth. Interestingly, when knocking down MMP2 only in the DP (*nubbin-Gal4*) we observed mild effects on bottom ECM thickness and disc bending (Supplementary Fig. 10j–m), suggesting that MMP2 is also required in the DP tissue to fine-tune ECM$_{DP}$ growth anisotropy and disc shape. In summary, we showed that spatial differences in MMP2 expression control the anisotropy of ECM growth thereby guiding *Drosophila* wing disc morphogenesis.

## Discussion

We investigated how growth of the wing imaginal disc affects its developing shape, namely tissue thickening and bending in a dome. Tissue doming emerges during growth and is associated with the build-up of elastic stress (Figs. 3 and 5). When this elastic energy is later relaxed by lysis of the PPE layer[52,53] and degradation of the ECM[51,54] during pupariation, the wing disc everts and gives rise to the wing blade. Thus, we propose that tissue doming pre-stresses the wing disc for future metamorphosis.

Our work sheds light on the mechanisms giving rise to tissue doming in response to growth-induced residual stress. Growth is intrinsically three-dimensional and requires a time-dependent measurement of cell and tissue volume. Here, we quantified growth at the cell and tissue scale in 3D over time and we find that it is not a growth differences between tissue layers that cause wing disc bending, but a growth differences between tissue and ECM. Previous studies have highlighted a role of local ECM degradation in the formation of epithelial folds[23] or that ECM layers can act as constraint for epithelial tissue expansion leading to cell and tissue shape deformations[31,32,34,35]. However, it has not previously been appreciated that the ECM which envelops the disc is not a passive boundary but an active, i.e., growing material. Growth and properties of the ECM can be actively modulated by the cell layers making the ECM a controllable boundary that mechanically feeds back on tissue shape as it grows.

We have studied the relative volume growth as well as the anisotropy of growth of both the tissue and ECM layers of the disc. We find that the two tissue layers follow planar growth, though the DP expands more in volume than the PPE leading to a growth differential between these two layers. Therefore, the PPE layer is stretched by the stronger growing DP layer providing an explanation for the decrease in PPE thickness during development (Fig. 1c'). However, this difference in tissue growth does not significantly impact the mechanics driving disc morphology since upon ECM digestion and MyoII inhibition disc doming is lost (Fig. 3b), suggesting that the thick DP layer mechanically dominates the much thinner PPE. Strikingly, we find that also the top ECM$_{PPE}$ layer follows planar growth that in magnitude is compatible with the mechanically dominating DP tissue layer. In contrast, the bottom ECM$_{DP}$ layer grows in 3D and its thickness increases constraining the expansion of the growing DP layer leading to stress accumulation and disc bending. Therefore, control over the growth anisotropy of the bottom ECM layer is key to explain morphogenesis of the wing disc.

We show that a nearly incompressible neo-Hookean model, in which pre-stress is added through differential growth anisotropy, can well recapitulate 3rd instar wing disc development. The assumption of incompressibility was based on measurements of ECM incompressibility in decellularization experiments (Supplementary Fig. 9c"). We use the neo-Hookean model as it is the simplest model of non-linear (finite strain) elasticity. But indeed, some ECMs can display strain-stiffening properties[55]. Generalised neo-Hookean models that capture this effect (Fung and Gent models) introduce a strain-hardening parameter which requires careful extension tests of the bottom ECM that were beyond the scope of this study.

We aimed to keep the number of model parameters small and made several simplifying assumptions. Firstly, we decided not to model the hinge region as it grows at similar rates as the wing pouch (Supplementary Fig. 3). We also assumed that the wing disc is axisymmetric and has a no-traction (i.e., force-free) boundary condition at the disc edge. This simplifying assumption is more correct in anterior–posterior direction than in dorsal–ventral direction, where complex folds form in larger discs[56]. Capturing the whole morphology of the disc will be an interesting challenge for future studies. Nevertheless, our modelling choices, guided by relative simplicity, were sufficient to understand key physical experiments like collagenase and decellularization results and to capture quantitatively key geometric quantities and predict the degree of growth anisotropy and stiffness ratio between the layers.

In this manuscript, we considered all layers to exhibit polar isotropy of the growth tensor, meaning the radial and orthoradial components of the growth tensor are identical. We note that we showed previously[37] that cell divisions in the wing disc exhibit a preferential orientation, suggesting some presence of polar anisotropy. However, its effect on tissue shape is likely negligible in the light of the presented collagenase experiments: Since no thickness gradients or bending were observed in the tissue after ECM digestion via collagenase, we conclude that any polar growth anisotropy in the tissue layer is not strong enough to cause significant tissue bending or thickening. Similarly, in light of the decellularization experiments, the relaxed configuration of the ECM layer is floppy and tensionless, showing that a strong enough pattern of polar growth anisotropy to alter the shape of the ten times thicker tissue layer is absent. We conclude that any polar growth anisotropy in the tissue layer and ECM layer is not strong enough to cause significant tissue bending or thickening, which is why we assumed polar isotropy of the growth tensor throughout this manuscript.

More complex versions of our model could provide insight in related biological questions such as the 3D shape remodelling process of disc eversion. Actomyosin contractility will be important to consider as supracellular cables in the periphery of the doming wing pouch[37] were proposed to form a constrictive ring required for disc evagination. It is possible that tissue bending induced by the ECM may induce actomyosin supracellular cables due to tissue stress accumulation and mechanosensitivity of actomyosin networks.

What controls the growth anisotropy of the ECM and what differs between the top and bottom ECM layers? Tissue-specific ECM biogenesis is complex and only starts to be revealed[57]. In *Drosophila* larvae the fat body is the main source for most ECM components, such as Collagen IV[47,58]. Therefore, the wing disc receives most ECM components via the haemolymph and their availability should not differ between the DP and the PPE. However, the disc tissue produces some portion of its Laminins[58,59], suggesting that local production of ECM components might control differences in ECM structure. Here, we provide evidence that local production of MMP2 in peripodial cells is required for planar growth of the top ECM layer. We suggest that MMP2 modulates ECM turnover, possibly by digesting Collagen IV, thereby increasing the viscous dissipation of elastic energy due to the growth of the underlying tissues. Therefore, the switch between planar versus thickness growth could be partially controlled by ECM turnover dynamics affecting ECM viscosity favouring either planar integration (high turnover) or thickness growth (low turnover). Future studies will address this interesting problem.

The ECM consists of a wide variety of molecules that influence their physical properties and response to stress[60]. Recently, the ECM component Hyaluronan was shown to induce osmotic ECM swelling during optic vesicle morphogenesis in zebrafish[61] and in chick pre-somitic mesoderm (PSM) elongation[62]. Swelling results in ECM expansion deforming the overlying epithelial layer or expanding the PSM. Therefore, differences in molecular composition determine the material properties and mechanical response of the ECM during development.

In this context, our study substantiates the emerging notion that the ECM is a complex material whose growth is actively regulated and required for morphogenesis.

## Methods

### Fly strains

The following fly lines were used: *y¹,w¹¹¹⁸, hs-Flp; act > Stop > Gal4, UAS-EGFP* (AyGAL4, originating from Bloomington stock 64231); *UAS-Histone::mRFP*[63], *Vkg^{G454}::GFP*[48] (both from F. Schnorrer), *vgQE-dsRed*[64], *sqh-Sqh::mCherry*[65] (insertion site 53B2), *endo-Ecad::GFP*[66], UAS-MMP2 (obtained from L. Le Goff, originally from the M. Grammont lab). The following lines were obtained from the Bloomington stock centre: *AGiR-Gal4* (#6773), UAS-PI3K_{DN} (#25918), UAS-CD8::mRFP (#27398), UAS-CD8::RFP (#27392), UAS-Mmp2TRiP (#61309), Mi{MIC} insertion in Mmp2 (#60512), *hh::Gal4* is described on FlyBase (www.flybase.org).

### Genotypes by figure

- Figure 1: *y¹,w¹¹¹⁸*
- Figure 2: (**c**) *hs-Flp; tub>Stop>Gal4, UAS-EGFP/UAS-Histone::mRFP*
- Figure 3: *Vkg^{G454}::GFP/endo-Ecad::GFP, sqh-Sqh::mCherry*
- Figure 4: *Vkg^{G454}::GFP/+*
- Figure 5: (**d**) *y¹,w¹¹¹⁸*
- Figure 6: (**a**) *y¹,w¹¹¹⁸*, (**b**) *w; ; hh-Gal4/ UAS-CD8::RFP*, (**c**) *w; UAS-Mmp2TRiP/+; hh-Gal4/UAS-CD8::RFP*, (**f, g**) *w; +/Vkg::GFP; hh-Gal4/UAS-CD8::RFP*, (**f, h**) *w; UAS-Mmp2TRiP/Vkg::GFP; hh-Gal4/UAS-CD8::RFP*
- Supplementary Fig. 1: *y¹,w¹¹¹⁸*
- Supplementary Fig. 2: *hs-Flp; tub>Stop>Gal4, UAS-EGFP/UAS-Histone::mRFP*
- Supplementary Fig. 3: (**a, c–f**) *hs-Flp; tub>Stop>Gal4, UAS-EGFP/ UAS-Histone::mRFP*, (**b**) *y¹,w¹¹¹⁸*
- Supplementary Fig. 4: **a** *y¹,w¹¹¹⁸*, **b, f** *UAS-CD8::mRFP/+; AGiR-Gal4/ +*, **c, g** *UAS-CD8::mRFP / UAS-PI3K_{DN} ; AGiR-Gal4/+*
- Supplementary Fig. 5: **a, c** *Vkg^{G454}::GFP/endo-Ecad::GFP, sqh-Sqh::mCherry*
- Supplementary Fig. 6: *w; UAS-GFP/+; UAS-MMP2/hh-Gal4, tub-Gal80ts*
- Supplementary Fig. 7: *Vkg^{G454}::GFP/+*
- Supplementary Fig. 8: *Vkg^{G454}::GFP/+*
- Supplementary Fig. 9: *Vkg^{G454}::GFP/+*
- Supplementary Fig. 10: (**a**) *Mi{MIC}* insertion in MMP2 (#60512), (**b, g**) *w; + ; hh-Gal4/UAS-CD8::RFP*, (**c, h**) *w; UAS-Mmp2TRiP/+ ; hh-Gal4/UAS-CD8::RFP*, (**d**) *w; UAS-Mmp2TRiP/Vkg::GFP; hh-Gal4/ UAS-CD8::RFP*, (**j**) *w; nub-Gal4/+; UAS-CD8::RFP/+*, (**k**) *w; nub-Gal4/UAS-MMP2 TRiP; UAS-CD8::RFP/+*.

### Antibodies

Primary antibodies used were mouse-anti-Wingless (4D4-s; 1:120; DSHB, University of Iowa); mouse-anti-Patched (Apa1-s; 1:40; DSHB, University of Iowa); rat-anti-DE-cadherin (DCAD2 concentrate; 1:200; DSHB, University of Iowa); rabbit-anti-GFP (1:1000, Abcam ab6556, Lot:GR3404234-1); rabbit-anti-Phospho-Histone H3 (PHH3, 1:1000, Cell Signaling #9701); rabbit-anti-Mmp2 (1:500, from ref. [67]); rabbit-anti-Vkg (1/500, from ref. [68]).

Tissue outlines were marked by Alexa Fluor 660 Phalloidin (1:100, A 22285, Sigma-Aldrich) which was added together with the other secondary antibodies. Secondary antibodies used were Alexa 488 (donkey anti-mouse A21202, donkey anti-rabbit A21206), Alexa 568 (donkey anti-mouse A10037, donkey anti-rabbit A10042, goat anti-rat A11077) and Alexa 647 (donkey anti-rabbit A31573) from Invitrogen and Alexa 647 (donkey anti-mouse 715 605 151) from Jackson ImmunoResearch. Secondary antibodies were used at 1:500 dilution. Discs were blocked in 2% normal donkey serum (017-000-121, Jackson ImmunoResearch).

### Sample collection, immunostaining and imaging

For staged samples, embryos were collected for 2 h intervals as described before[69] and allowed to develop at 25 °C until the desired developmental stage (MMP2 knockdown experiments in Fig. 6b–i were performed at 29 °C due to increased efficiency of knockdown). Wing discs were isolated at defined time intervals after egg laying (hAEL). For 72hAEL and older time points, only male larvae were included (positive selection by the transparent genitalia disc well visible in the posterior half of male larvae); 65hAEL data contains male and female larvae since at this time point the genitalia disc is not yet clearly visible.

All larvae of one experiment were dissected, processed and imaged in parallel, using identical solutions in order to reduce experimental variations. Immunostaining of imaginal discs was performed as described previously[69]. Discs were mounted in Vectashield Plus (H-1900, Vector Laboratories) using double-sided tape as spacers (TESA 05338) to maintain tissue morphology and avoid squishing of the sample.

All fixed samples were imaged on a Leica SP8 confocal microscope (Leica Application Suite X 3.55.19976) using a ×40 or a ×63/1.4 NA oil-immersion objective. All image stacks of one experiment were acquired in the same session using identical imaging settings. Imaging conditions were chosen to be well within the dynamic range of the fluorescent signal obtained. For optical cross-sections of wing discs stacks with high resolution along the *z* axis were acquired (typically 0.33 μm of spacing between slices).

### Image processing

Image data were processed and quantified using Fiji/ImageJ software (version 1.53t, National Institute of Health). Further data processing was performed in Python 3.9.7 (Anaconda Navigator 2.1.1, Spyder 5.1.5, Seaborn 0.11.2). Individual procedures are described in detail in the following:

**Epithelial thickness and bending quantifications.** Image stacks of high resolution along the *z* axis of discs stained for Wingless (Wg) and Patched (Ptc) were acquired (typical spacing between slices are 0.33 μm). Stacks were sliced using the 'Reslice [/]' function in Fiji to obtain optical cross-sections parallel to the dorsal/ventral boundary with a slight dorsal offset. The thickness of the disc proper layer and the overlaying peripodial layer were measured at the position of the cross marked by the horizontal Wg and the vertical Ptc expression using the 'Straight line' tool.

In order to visualise the average basal shape of the disc proper epithelium, the basal outline of the disc proper epithelium was marked using the 'Kappa−Curvature Analysis' plugin in Fiji. Kappa allows the export of a spline-fit of the basal surface outline. Basal outlines were registered along the *x* axis, defining 0 as the position of the A/P boundary (marked by Ptc). Registered profiles were subsequently fitted in Python by a B-spline using a custom script. In plots, the average basal outline is depicted by a solid red line and profiles of individual discs in grey.

**3D segmentation and volume qualifications (Supplementary Fig. 1).** In order to assess volume growth at the tissue level we have focused on the wing pouch in the DP epithelium. We have used antibody stainings against Wingless (Wg) and Patched (Ptc) to mark the pouch by the inner ring of Wg expression. We have followed two approaches to obtain volume information of the wing pouch:

(1) First, we have performed proper 3D segmentation of volumetric image stacks of staged wing discs (72 and 118hAEL) stained for Wg/Ptc (see Supplementary Fig. 1e–g). The Wg/Ptc staining results in sufficient labelling of epithelial outlines. We segmented the epithelial volume (Wg/Ptc signal) in Ilastik 1.3.3[70] using the 'pixel classification' and obtaining a binary mask of the segmented epithelial signal. The binary mask was manually corrected for errors in Fiji and finally restricted to the volume surrounded by the inner Wg ring. Volume values for the segmented wing pouch were obtained from the binary mask using the 'Histogram' function and by multiplying the obtained number of wing pouch pixels with the voxel volume. Indeed, this procedure is work intensive since it requires a significant amount of manual correction in Fiji.

(2)     We therefore have tried to approximate the wing pouch volume by simply multiplying pouch epithelial thickness with the area of the inner Wg ring (see Supplementary Fig. 1a–d). Epithelial thickness was measured close to the intersection of the Wg/Ptc cross. Wg ring area was measured in maximum projections using the 'Polygon selection' tool in Fiji. In order to correct for the increased area due to tissue doming after 80hAEL we have approximated the Wg ring area as spherical cap with cap height $h$ (which was obtained from the average basal outlines in Fig. 1b'). Please see Supplementary Fig. 1b, c for details on this correction. Indeed, this approximation yields values very close to the 3D-segmented 'true' values for both the flat 72hAEL and the bend 118hAEL time points (see Supplementary Fig. 1g). We therefore used the less work intense approximation approach to quantify wing pouch volume in Fig. 1d'.

Analogous to procedure (1), we segmented the volume of the peripodial epithelium overlying the wing pouch (see Supplementary Fig. 1h–j). Due to a lack of landmarks in the peripodial layer, we have decided to quantify the volume of the peripodial tissue that covers the inner Wg ring. Hence peripodial volume values plotted in Supplementary Fig. 1j correspond to the peripodial volume that covers the wing pouch tissue.

**Quantification of clonal proliferation rates (Supplementary Fig. 3a).** Clones were induced by heat-shock-induced cassette recombination that resulted in clonal expression of EGFP and Histone::RFP (*hs-Flp; act > Stop > Gal4, UAS-EGFP/UAS-Histone::RFP*). As a general rule, wing discs were isolated 24 h after clone induction, therefore staged larvae were heat shocked (HS) at 37 °C at defined time points and dissected 24 h later (hence HS at 48hAEL for 72hAEL samples). HS length was shorter for early time points (4 min for 72hAEL sample) and longer for older samples (7 min for 116hAEL) to obtain discs with sparse clonal density in order to reduce clone fusion and associated mistakes in estimating clonal proliferation rates. Isolated discs containing clones were stained for GFP, RFP and Wg/Ptc (landmarks) and imaged using a Leica SP8 confocal microscope at ×63 magnification. Nuclei per clone were counted using the Fiji '3D viewer' and the 'orthogonal views' tools in order to correctly assess nuclear numbers in 3D. Given that each clone originates from a single founder cell, the cell number $n$ after 24 h of clone induction allows us to calculate the number of proliferation events that have taken place within these 24 h using $\log_2(n)$. Average spatial proliferation maps were created by arranging the data according to the landmarks provided by the Wg/Ptc staining.

**Quantification of proliferation rates via PHH3 (Supplementary Fig. 3b).** Wing discs ($y^1,w^{1118}$) were isolated at defined time points of development and stained for Phospho-Histone H3 (PHH3, a marker for mitosis), E-cadherin (cell outlines), and Wg/Ptc (serving as landmarks for registration of multiple discs). Image stacks were obtained on a Leica SP8 microscope using a ×63 objective. Apical surface projections of the disc proper surface using the E-Cad signal were obtained using a custom-made Fiji plugin based on the 'Stack Focuser' plugin. Subsequently, cell outlines were segmented using the 'Tissue Analyzer' plugin[71]. The landmarks provided by the Wg/Ptc cross and ring were used to register multiple discs and to obtain average spatial distributions of cell area and cell density for each time class. As published previously[36], cell area becomes non-uniform around 80hAEL with smaller cells in the centre compared to the periphery of the wing pouch. Average proliferation profiles were created based on the PHH3 signal. Given that cell density is not uniform in space, we subsequently binned the cell density and proliferation density profiles in rectangular regions of 8 μm edge length. Proliferation density profiles were normalised per bin to obtain the average proliferation rate per cell in a spatial manner. In order to investigate spatial non-uniformities

in proliferation between the centre and the periphery of the disc we divided the wing disc in 4 elliptic rings within the region marked by the inner Wg ring. While the central 3 regions correspond to wing pouch tissue, the outermost region corresponds to hinge tissue (see Supplementary Fig. 3b, bottom). Importantly, independent of the method chosen for data quantification, we never obtained higher proliferation values in the centre versus the periphery. In contrast, correct normalisation of cell proliferation by cell density shows a tendency of decreased central proliferation after 80hAEL.

**Extraction of Vkg::GFP concentration profiles and ECM thickness.** We have used the Vkg::GFP signal to assess changes in ECM structure and thickness. In order to quantify absolute changes in Vkg::GFP levels and distribution we have acquired image stacks with high z-resolution of either the top $ECM_{PPE}$ or the bottom $ECM_{DP}$. Importantly, we imaged either the top or bottom ECM depending on the orientation of the wing disc and which ECM (DP or PPE) was closer to the objective after mounting. Optical cross-sections of image stacks close to the Wg/Ptc intersection were obtained using the 'reslice' function in Fiji. From average projections of five consecutive slices we extracted the Vkg::GFP profiles using the 'straight line tool' (3.6 μm width) and the 'plot profile' function in Fiji at three random positions (in order to average out small local differences). These three profiles were aligned by the position of their peak intensity and averaged. Finally, average profiles from multiple wing discs were averaged in order to obtain representative Vkg::GFP profiles for the top $ECM_{PPE}$ and the bottom $ECM_{DP}$ for the different developmental time points (see Supplementary Fig. 7b', c'). Profiles were plotted in Python 3.9.7/Spyder 5.1.5 using the Seaborn package 0.11.2 ('lineplot' command). In the average profiles the dashed line indicated the average fluorescent intensity and the error bands the standard deviation.

In order to quantify changes in ECM thickness we chose intensity thresholds at values that capture most of the observed Vkg::GFP fluorescence (see Supplementary Fig. 7b', c', threshold values = 10a.u.). We measured the width of the Vkg::GFP profiles at the given threshold in order to compare changes in thickness between different time classes and experimental treatments (see Supplementary Fig. 7b", c"). Analogously, we extracted profiles in ex vivo cultured discs upon decellularization (see Supplementary Fig. 9c, d). The only difference was the chosen intensity threshold values (10 a.u. for the $ECM_{DP}$ and 15a.u. for the $ECM_{PPE}$) that differed due to the different imaging conditions of the live sample.

**Details on ex vivo culture and imaging**
The procedure of long-term imaging of disc explants was based on the protocol published in ref. [72] with minor modifications. We slightly modified the composition of the culture medium by adding adenosine deaminase (ADA, 8.3 ng/ml final concentration, Roche 10102105001) as proposed by recent findings of ref. [73]. In our hands, the addition of ADA in particular improved the long-term culture of young disc explants. In contrast, the addition of juvenile hormone (Methoprene) as proposed by Strassburger et al. has not proven beneficial in our setting and we did not use it in our culture medium.

As described previously[72], 72hAEL old larvae were dissected in culture medium and explants were immobilised between a round coverslip and a porous filter membrane (Whatman cyclopore polycarbonate membranes; Sigma, WHA70602513) using double-sided tape as spacers (-50 μm thickness, 3 M Scotch ATG 904 Clear Transfer Tape, No. 909-3799 from RS Components). The coverslip containing the mounted explants was inserted in an Attofluor chamber (A7816, ThermoFisher) and filled with 1 ml of culture medium. Explants were imaged on a Nikon Roper spinning disc Eclipse Ti inverted microscope (controlled by Metamorph 7.8.4.0) using a 40×−1.25 N.A. water-immersion objective at 22 °C. Image stacks of 1 μm z-spacing were acquired in 10 min intervals for up to 12 h.

**Quantification of volume growth rates.** We staged larvae of the genotype *hs-Flp; act > Stop > Gal4, UAS-EGFP* to 72hAEL. Clonal expression of a cytosolic GFP was induced by heat shock (at 37 °C for 4 min) 12 h before dissection (60hAEL). Wing discs of 72hAEL old animals were isolated, cultured and imaged as described before. Individual clones from volumetric movies were cropped in Fiji, the background was subtracted (using a 'rolling ball' radius of 50 pixel) and the volume marked by the cytosolic GFP signal segmented using a 'pixel classification' in Ilastik[70]. Clonal volumes were assessed for each hour of the movie by averaging three consecutive time points and the hourly growth rate was calculated (see plots in Supplementary Fig. 3c'). For each clone, we calculated an average growth rate for the full span of the movie (up to 10 h). In order to compare spatial differences in growth rates in the plane of the DP epithelium we grouped central clones, defined as clones within an ellipse covering the central 60% of the wing discs (see scheme Supplementary Fig. 3d) and compared their growth rate to peripheral ones. Analogously, we segmented and analysed growth rates in the peripodial epithelium.

### Correlating relative tissue with ECM growth (Supplementary Figs. 7 and 8)

In order to gain understanding of the volumetric increase of the epithelial compared to the ECM layers we plotted the relative volume increase of the tissue versus the relative increase in the ECM layer. For the DP, volume was quantified in Fig. 1d', and normalised by the average value either at 65hAEL or at 72hAEL. In the time interval between 65 and 118hAEL, the DP volume increases by ~65.8-fold and between 72 and 118hAEL by ~19.9-fold. For the bottom $ECM_{DP}$ layer we first estimated the volume by multiplication of the known area of the inner Wg ring (see Supplementary Fig. 1) with the known $ECM_{DP}$ thickness (Supplementary Fig. 7c"). Estimated ECM volume increases ~16.2-fold from 72 to 118hAEL.

In addition, we quantified Viking::GFP integrated intensities of the signal lining the basal side of the wing pouch (area within the inner Wg ring). For this we created a temporal data set of disc of the genotype Vkg::GFP/+. Processing (fixation, immunostaining and mounting) was done under identical conditions using identical solutions. Subsequently, the mounted discs were imaged in one session using identical settings to allow direct comparison of fluorescent Vkg::GFP intensities. Only discs with their basal side of the DP facing towards the objective were imaged and included in the quantifications. For processing (in Fiji), the Vkg::GFP fluorescent signal was restricted to the volume lining the inner Wg ring and after background subtraction (rolling ball with radius = 50) the integrated Vkg::GFP intensity was calculated using the 'histogram' function. We found that the integrated fluorescence intensity of the Vkg::GFP marked $ECM_{DP}$ increases by ~56.6-fold between 65 and 118hAEL. Consistently, both approaches, ECM volume estimation and integrated intensity measurements, suggest that the volumetric growth of the ECM is reduced compared to the overlaying DP tissue.

We performed the analogous analysis for the peripodial layer. Given the lack of landmarks in the PPE layer, we decided to include the peripodial volume that overlays the inner Wg ring (corresponding to the wing pouch) in our quantifications. The peripodial volume overlaying the Wg ring increased by ~8.2-fold between 72 and 118hAEL (see Supplementary Fig. 1h–k). The integrated $Vkg::GFP_{PPE}$ intensity in the top $ECM_{PPE}$ (see Supplementary Fig. 8a, b) increased by ~19.3-fold between 72 and 118hAEL, a value very similar to the volume increase of the DP layer (~19.9-fold).

### Acute ECM modifications and decellularization

**Acute ECM digestion (Fig. 3 and Supplementary Fig. 5).** Larvae of the genotype $Vkg^{G454}::GFP/endo-Ecad::GFP$, *sqh-Sqh::mCherry* were staged as described above. Larvae were dissected in PBS and transferred to Eppendorf tubes containing PBS on a 37 °C heat block. In order to inhibit Myosin II activity the ROCK inhibitor Y-27632 dihydrochloride (Sigma-Aldrich, Y0503) was added to a final concentration of 2 mM and incubated for 2 min. Subsequently, the extracellular matrix was digested by the addition of Collagenase (Sigma-Aldrich, C0130) at a final concentration of 1 mg/ml and incubated for 1 min. After 1 min of Collagenase treatment, discs were fixed by direct addition of fixative (4% PFA in PBS) to maintain and conserve disc morphology after ECM digestion. Discs were fixed for 20 min at RT on a rocker and subsequently processed for immunostaining as described above.

**Decellularization of wing discs (Fig. 4 and Supplementary Fig. 9a, b).** Here, we have adopted and used a chemical decellularization method to free the wing disc extracellular matrix from the load exerted by the epithelial cell layers. While classical decellularization protocols often use strong detergents like e.g., SDS, wing disc cells are soft and increased concentrations of Triton X-100 are sufficient to permeabilize and degrade cells. Chemical decellularization was used in various regenerative[74–76] and biomedical approaches[77] and decellularization by Triton X-100 retains ECM microstructure[78].

$Vkg^{454}::GFP$ larvae were staged to 72, 96 and 118hAEL to cover the whole 3rd instar development. For decellularization, in PBS dissected and inverted larvae were incubated in PBS + 3% Triton X-100 for 15 min before fixation, while control discs were incubated in PBS. Shorter exposure to Triton X-100 did not result in sufficient separation of the ECM layer from disc proper cells; more than 15 min resulted in a loss of the cell layer and hence a loss of the required landmarks to identify peripodial versus disc proper ECM layers. After fixation, discs were processed for immunofluorescence as described before.

All discs were mounted on the same microscopy slide and imaged under identical conditions. For each wing disc, depending on its orientation, the ECM closer to the objective was imaged (either peripodial or disc proper ECM). Hence, only image data acquired close to the objective was included in intensity quantifications to avoid inaccuracy due to loss of signal with increasing imaging depth.

Optical cross-sections of the region around the intersection of the A/P D/V boundaries were obtained by slicing the image stack using the 'reslice function' in Fiji/ImageJ. Intensity profiles along the apical–basal axis of the Vkg::GFP signal were obtained using the 'straight line tool' (line width of 3.6 μm). Multiple profiles were aligned according to their peak intensity, averaged and plotted in Python using the Seaborn library 0.11.2 (lineplot function, error bands represent the standard deviation). Thickness changes of the ECM layer under load and upon relaxation were quantified as described in the section 'Extraction of Vkg::GFP concentration profiles and ECM thickness'.

**Circular bleaching and ex vivo decellularization (Fig. 4d and Supplementary Fig. 9c, d).** Decellularization results in a loss of the epithelial layers and hence landmarks provided by the cell layers (like e.g., the Wg/Ptc ring and cross). In order to assess relaxation in the *x/y*-plane of the ECM we used a photobleaching approach to mark circular regions that could serve as traceable landmarks that are not lost during decellularization.

Vkg::GFP wing discs of 116hAEL were isolated in PBS and glued to the bottom of a petri dish using classical embryo glue. Embryo glue was applied shortly before mounting, briefly allowed to dry and then covered by a drop of PBS in which the discs were arranged. For experiments assessing the $ECM_{DP}$, discs were glued with their PPE side to the bottom of the petri dish, their $ECM_{DP}$ facing upwards. For assessing the $ECM_{PPE}$ discs were mounted with inverse orientation.

Experiments were performed on an upright Nikon A1R MP + multiphoton microscope (controlled by Nikon NIS-Elements AR 5.11.01 software). Mounted live discs were taken directly to the microscope and imaged from the top using a water-immersion objective (40x/1.15NA). For the excitation of GFP a tuneable

wavelength pulsed laser (Coherent) at 920 nm was used. Imaging settings were optimised to use minimal laser power. Circular regions of interest (ROI, usually three circles per disc) were bleached using elevated laser power and scanning the ROI for 30 times. Depending on the geometry of the ECM this was repeated for multiple positions along the z axis to obtain a clear circle upon the maximum projection of an image stack (1 μm spacing). The circular bleaching procedure was restricted to a total of 40 min (~5 discs per session) such that including the mounting time the total time of discs in PBS did not exceed one hour before decellularization. After marking circles on all discs, the petri dish was filled with PBS containing 3% Triton-X-100 (PBST-3%). Discs were imaged before addition of Triton X-100 and subsequently in 10 min intervals after exposure to Triton X-100 in order to follow ECM shape changes due to the loss of constraints induced by the cell layers. Obtained image stacks were subsequently oriented in Fiji (using the reslice and transformation functions) to ensure that circles are not tilted but are in-plane with the projection plane. Circular area was measured in Fiji using the 'polygon selection tool' in maximum projections. Relative area changes were processed in Excel and plotted in Python (Seaborn library).

In order to investigate ECM thickness changes upon decellularization, samples were prepared the same way, however, per disc only two circles were bleached and image stacks with high z-resolution were obtained (0.25 μm spacing). Sixty minutes after exposure to Triton X-100, another set of high z-resolution image stacks were acquired under identical settings. Subsequently, ECM thickness profiles were processed as described in the section 'Extraction of Vkg::GFP concentration profiles and ECM thickness'.

### Quantification of changes in peripodial cell architecture upon MMP2KD (Fig. 6b–e)

The peripodial layer is thinner and shows a lower fluorescent signal (of e.g., Ecad or RFP) than the DP layer. Even after optimisation of the Stack Focuser plugin, we were not able to obtain satisfiable results for apical surface projections of the peripodial layer. We therefore manually create a mask for the junctional plane of the PPE layer. This was done in Fiji creating an additional channel ('mask$_{PPE}$') in which the peripodial surface was marked using the pencil tool such that in the final mask$_{PPE}$ stack pixels either had a value of 1 if they correspond to the peripodial apical surface or a value of 0 otherwise. Multiplying the mask stack with the Ecad stack (using the 'image calculator' function) allowed us to extract only peripodial Ecad signal which after maximum projection yielded the PPE junctional plane. Cell outlines in PPE projections were then segmented in the region of the A/P boundary using Tissue Analyzer[71].

Epithelial thickness was measured in cross-sections obtained by using the 'reslice' tool in Fiji. Thickness was measured using the 'straight line' tool (Fiji) 10 μm posterior of the peripodial A/P boundary (as indicated in Fig. 6b', c'). In the same cross-sections, the distance between the peripodial and the disc proper A/P boundary was measured using the 'segmented line' tool.

### Statistics and data representation

Given the experimental constraints we aimed to obtain a sample size large enough ($n \geq 5$) to allow testing statistical significance by using a two-sided Student's $t$ test (unequal variance, *$P \leq 0.05$, **$P \leq 0.005$, ***$P \leq 0.0005$). The number of samples and $P$ values are either indicated in the figure or the respective legend. For each experiment, $n$-numbers indicate biological replicates, meaning the number of biological specimens evaluated (e.g., the number of wing discs or clones). Plots were created in Python using the Seaborn library. In line plots, the error bands indicate the standard deviation. In box plots, the median is indicated by a central thick line while the interquartile range (containing 50% of

the data points) is outlined by a box. Whiskers indicate the minimum and maximum data range, outliers are indicated by a black rhomb and were excluded from further processing.

### Reporting summary

Further information on research design is available in the Nature Portfolio Reporting Summary linked to this article.

## Data availability

The data supporting the findings of this study and material are available on request from the corresponding author (T.L.). Source data are provided with this paper.

## Code availability

A supporting text describing the modelling and fitting procedures is available in the supplementary information. The code used for data analysis, simulations, and model fitting is available from the corresponding author (T.L.) on request.

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

## Acknowledgements

We would like to thank all the members of the Lecuit team for stimulating discussions. We also thank Larry Taber (Washington University in St. Louis) for instructions and sample codes in the implementation of growth in a finite element framework. We are grateful to B. Aigouy (IBDM, France) for help with Tissue Analyzer,Benoit Dehapiot (CENTURI, France) for help with Python and to Stephane Noselli (IBV, France), Frank Schnorrer (IBDM, France), Kendal Broadie (Vanderbilt, USA), Loïc Le Goff (Institut Fresnel, France), the Bloomington Stock Center and the Developmental Studies Hybridoma Bank (University of Iowa, USA) for flies and reagents. The IBDM imaging platform and the France-BioImaging infrastructure supported by the Agence Nationale de la Recherche (ANR-10-INSB-04-01; call "Investissements d'Avenir") provided support. This work was supported by the Ligue Nationale Contre le Cancer (Equipe Labellisée 2018). S.H. was supported by an EMBO long-term fellowship (ALTF 217-2017), the College de France (Paris, France). S.H and A.E. were supported by postdoctoral fellowships from the Turing Center for Living Systems (CENTURI), funded by France 2030, the French Government programme managed by the French National Research Agency (ANR-16-CONV-0001) and from Excellence Initiative of Aix-Marseille University - A*MIDEX ».

## Author contributions

S.H., A.E., G.Z., C.E. and T.L. conceived and designed the study. S.H. performed the experiments and quantified the data. A.E. designed the computational model and performed the simulations. All authors discussed the data. S.H., A.E. and T.L. wrote the manuscript. All authors read and approved the manuscript.

## Competing interests

The authors declare no competing interests.
