## [Peer Review File · Nature Communications]

Growth anisotropy of the extracellular matrix shapes a developing organREVIEWER COMMENTS

Reviewer #1 (Remarks to the Author):

"p" refers to page number and "L" refers to line number

* General comments

This study investigate the morphogenesis of the drosophila wing imaginal disc which undergoes both volumetric growth and bending during its development.

While the most natural hypotheses to explain the doming of growing quasi-planar tissues are either planar differential or planar anisotropic growth, several experiments on the growth rates of the different cell layers composing the imaginal disc are presented and used to motivate the need for a multilayered model based on morpho-elasticity.

The theoretical model is correct. The word "predict" occurs 8 times and it seems an overclaim. In fact, the results of the model correctly "describe" the experiment. The model shows that a description with three layers which grow differently reproduces the experimental observations, while the simpler versions tested by the authors fail to reproduce the experiments.

The authors should explicitly indicate in the introduction that they do not look for parsimony and that the complexity of their model ingredients is justified by the agreement with observations. In fact, one can imagine other simple models which could have been tested. For example a single ECM layer growing differentially : space-dependent/anisotropic planar growth combined with an appropriate homogeneous component of the growth tensor in the Z direction. Or maybe a two layers system incorporating similar features. Since neither data on the spatial differences (in the plane) of the growth rates of the ECM, nor any data on the degree of planar anisotropy (i.e differences between radial and orthoradial growth components) of any layer are presented, such simpler models unfortunately cannot be ruled out by this study in its present form.

* Main text

Experiments are careful. This part of the text is clear and written with many details.

There should be one short sentence, aimed at non-specialists, to summarize the exact mechanism for bending, according to the authors. Or make an analogy with daily life objects.

If (x,y) is the plane of the tissue in cartesian coordinates, (r,θ) the same plane in polar coordinates, and z the perpendicular direction, "anisotropic growth" can either mean :

- growth along x is different from along y : elongation, or extension compression
 - growth along r is different from along θ : polar anisotropic growth
 - growth along Z is different from along x,y : in-plane growth differs from out-of-plane thickening
- In the latter case, which is extensively studied in this manuscript, it might be better to avoid the words "anisotropy" or "anisotropic growth" which are an unnecessary source of confusion.

- p2 L11 :

It would be useful to mention here the effect of differential growth in plants (including flowers, see eg the papers of Enrico Coen).

Some parts of the Discussion (e.g. p11 L25) could be moved to Introduction, where differential growth in general is presented.

The Discussion should rather focus on the perspectives opened by the current model, which are much more specific.

- section "Tissue thickening is not due to differential cell growth in the tissue plane" :

In this section, the hypothesis that planar differential growth drives tissue thickening is investigated. It is shown that, using a crude space-dependent growth tensor consisting of an inner fast growing core surrounded by an annulus growing at a slower pace, the observed thickening cannot be reproduced with reasonable growth rates differences between inner and outer sections. While this is interesting, the most natural hypothesis for the thickening of the tissue is simply a homogeneous growth process in the "Z" direction. Although this natural hypothesis is mentioned, and ruled out, later (in the "The ECM is essential for tissue bending" section) it would be better to discuss this hypothesis right at the beginning of this section.

- section "Bending is not due to differential growth of the tissue layers" :

Although planar differential growth (spatial heterogeneity, see Klein et al, Science 315:5815 (2007)) can lead to a pronounced doming of a single layer, it should be noted that planar growth anisotropic in polar coordinates can also lead to a pronounced "doming" of a single growing layer (or for two layers growing at the same rate), see Dervaux and Ben Amar, PRL 101, 068101 (2008). This occurs for example in the algae acetabularia acetabulum where the degree of anisotropy controls the tissue curvature. Could the authors measure this anisotropy in their datasets or at least discuss this point ? Anisotropy is defined here as a difference between the radial and orthoradial components of the growth tensor.

- section "Discussion"

Perspectives should be discussed more precisely. In principle, could the present study be extended to describe the fact that the wing disc is eventually fully folded (p10 L13 or p10 L29, explain what additional ingredients would be required) ?

* Figures

- Fig 1b, p16 L8 :

recall here the definition of hAEL (currently it is only defined at p3 L7).

* Methods

- p26 L10 : grammar

* Extended Data Figures

- p43-44 : Fig 6d should be in chronological order, unless there is a good reason for antichronological one.

* Supplementary information / Elastic model :

The authors should re-read it to bring it to the level and clarity of the main text.

This part seems to be written much more hastily than the main text :

- inconsistent notations (both J_A and Δ) are used to denote the determinant of the elastic part of the deformation tensor),
- alternative definitions (r and θ can indifferently be called either "polar" or "cylindrical" coordinates but not both),
- unclear or allusive sentences, in particular (but not only) the paragraph "With the parameters" (p4)
- equations lacking explanations,
- several typos, spurious or missing spaces,
- low legibility figures (check font size),
- lack of transition between two paragraphs,
- calling figures in main text and in Supp Info with the same numbers, resulting in confusion,
- etc.

- section 1.2.1. of the SI Elastic model:

The conclusion at the end of this section is surprising and seems to contradict both analytical and experimental results. It was demonstrated that either anisotropic (in cylindrical coordinates) or isotropic but space-dependent growth leads to a pronounced "doming" of a single growing layer. The results shown in this section, which lead to a different conclusion, might be a consequence of the peculiar growth tensor (consisting of an inner fast growing core surrounded by an annulus growing at a slower pace) chosen here. But this is not a general conclusion and cannot motivate the need for a more sophisticated model.

Surely the authors do not mean that doming cannot be reproduced with a space-dependent tensor (which is likely incorrect) but rather that the measured thickening of the system cannot be reproduced with a simple space-dependent planar growth tensor.

In the same section, it is also noted that the softness of the material excludes a doming. This is surprising since, in the limit of an incompressible material made of a single layer, the final shape of the growing layer should be independent of the stiffness of the layer (the stiffness is only a prefactor which fixes the magnitude of the stresses). Can you also clarify this point ?

Stress free situations can have different causes. The legend of Fig 2b states that the grown system is stress free if the different layers grows at the same rate. But this is obviously not true, in all generality, if the growth tensor is either anisotropic (in the plane) or space-dependent. The caption does not specify whether the growth is heterogeneous or homogeneous; on the picture the growth is apparently heterogeneous. Can you also clarify this point ?

- The sentences before and after Eq. (33) should be reconsidered. It should be mentioned that Ref. [5] is an anterior work of one of the authors. Hence, if the authors have noticed a mistake, they should first publish an erratum (the procedure is simple and quick), and then quote here both Ref. [5] and its erratum. This is a correct scientific practice.

Reviewer #2 (Remarks to the Author):

Harmansa et al, investigate the role of differential growth within epithelia and across the epithelial and extracellular matrix layers in determining the final domed morphology of the wing imaginal disc. Combining a large experimental dataset with quantitative measurements and computational modeling, they propose that the domed shape of the wing pouch can't be explained by differential growth rates within the tissue layer, but by differential growth between the epithelia and the basal extracellular matrix. The role of local ECM regulation in wing disc growth and morphogenesis is already demonstrated by others. Therefore, the data need to convincingly establish that ECM growth anisotropy is actively controlling the final tissue shape. Several of the model assumptions are not very well supported and additional experimental data are needed to back up the authors' conclusions. While the authors' work provides new information to advance our understanding of the role of different tissue layers in epithelial morphogenesis, these issues must be addressed for the work to be recommended for publication.

Major points:

1. The authors model the growth of the tissue layers as in plane only. While it is well established that cell proliferation is indeed in-plane, this does not imply that growth take place in the plane. In addition, the authors' own data clearly show that there is substantial non-planar tissue growth (i.e. increase in thickness). It is not at all clear why the authors think the observed DP growth in z-axis is dispensable for their model. How would the model predictions change if the authors indeed taken DP growth in z into consideration?

Furthermore, the authors use experimental data on volume change rates to determine the γ_{DP} , while presuming $\gamma_Z = 1$ in their calculations. Knowing that thickness increase is a major contributor to the volume changes measured experimentally, this would significantly distort their estimation of γ_{DP} used in their model. Using measurements of area change over time would be a better estimate for γ_{DP} .

Again based on experimental data on volumetric growth, the authors use a γ_{ECM} that is lower than the γ_{DP} in their model. However, the area used in DP and ECM volume calculations is apparently the same throughout the study. Wouldn't this imply similar rates of planar growth for DP and ECM?

It appears that estimating planar growth rates based on volumetric growth data is inflating the difference between the DP and ECM planar growth rates used in the final model. Could this inflated planar growth difference itself, without the anisotropic growth difference, explain the doming effect? (i.e. would the scenario given in SI 1.2.4 predict the final tissue shape if the γ_{ECM} value used was not higher but same as the one used in SI 1.2.3?)

Overall, the choice of taking $\gamma_{ECM} \neq 1$ while keeping $\gamma_{DP} = 1$ is a key assumption of the model that is the bases of most of the authors final conclusions. However, this choice does not appear to be substantiated in the experimental data. This issue must be experimentally resolved.

2. The authors use cultured wing discs to show that clonal growth rates are consistent along the disc proper, and across disc layers. However, they do not establish that cultured discs go through the same morphological changes as discs in vivo. Indeed, for the DP clones that are shown in Figure 2c and

Extended Figure 3 e, we do not observe the thickness changes reported for the disc proper in Figure 1. To support the authors' conclusion that differential growth is not the cause of thickness increase and the domed shape of the disc, it should be shown that the discs can indeed go through thickening and doming *ex vivo*.

In addition, the timing chosen for the *ex vivo* experiments does not correspond to the time of maximal change in thickness or doming behavior. To strengthen their point, the authors should show clonal growth data from later time points (i.e. 96-118h).

Indeed, the results shown for growth rates in DP and PPE at 72h in Figure 2c' and authors conclusions on them, apparently contradict the authors' data shown in Extended Figure 2f and g, where the authors report a clear cellular volume increase for DP from 72 to 96h while the PPE volume remains unchanged. How do the authors reconcile this contradiction? The lack of PPE cell volume change from 72h to 96h AEL (Extended figure 2g) is also in apparent disagreement of the continuous PPE volume increase from 72-118h AEL at the tissue scale (Extended Figure 1j). The authors should comment on their interpretation of these differences and justify their choice of using the clonal growth rates measured *ex vivo* at 72h to verify their model.

Could the authors use the differences in volume change rate observed at the tissue scale over the time of disc development to verify their model? Considering the limitations of the *ex vivo* system used to measure clonal growth rates, the tissue scale measurements done *in vivo* would be more accurate and more relevant.

3. The authors block proliferation in PPE to show that a higher growth rate of the PPE is not the cause of disc doming. However, by doing so, they presumably create a growth differential in the other direction (i.e. the DP now has a higher growth rate than the PPE). According to the authors' model, wouldn't this differential result in bending of the disc in the other direction? One explanation is that, in response to growth inhibition in the PPE, the growth of DP is altered as well (as is suggested by the apparently smaller overall size of the disc shown in Extended Figure 4d). The authors should check if this is the case, and demonstrate that this does not impact the interpretation of their results.

4. The authors estimate PPE ECM volume by multiplying disc area by ECM thickness. However, as the authors note, the image in Extended Figure 6d show significant wrinkling of the ECM layer at 118h AEL. This suggest an increase in ECM volume (i.e. growth) that is not taken into consideration in the growth measurements. As the similar growth rate of the PPE and DP tissues and the PPE ECM is one of the major aspects of their model and their findings, the authors should account for the ECM growth that is represented in the PPE ECM wrinkling. Could the authors use Vkg intensity to measure growth of PPE ECM as they have done for DP ECM?

5. The authors conclude that the difference in growth anisotropy is key in explaining the domed morphology of the disc. The increase in ECM thickness would suggest an increase in stiffness as well. Could the authors exclude a role of ECM stiffness?

6. The authors favor a role of ECM over contractility in the control of bending based on acute inhibition of contractility and ECM degradation. Yet these results could be interpreted as a function of ECM in the maintenance of DP shape. The authors need to characterize the apical and basal actomyosin distributions during DP growth and analyze the impact of MyoII inhibition on clone and tissue growth. A detailed description of the previous findings on the role of ECM and actomyosin contractility (Nematbakhsh et al, 2020) need to be provided.

7. The DP bending and the DP shape are not quantitatively compared with the ones obtained by modelling (Fig. 5 and Extended Data Fig 6h). Quantitative comparisons of the DP bending *in vivo* and

in silico in the different experimental conditions is needed to validate the authors' conclusions.

8. The authors provide perturbation of ECM growth in the PPE will lead to tissue bending. However, they do not demonstrate that introducing ECM anisotropic growth in the DP is necessary to promote dome shape formation. The author needs to directly test the role of ECM growth anisotropy in the DP to substantiate their conclusions.

9. The authors write "These studies suggest a universal mechanism where a passive ECM layer constrains epithelial growth and controls organ morphogenesis. Here, we demonstrate that during *Drosophila* wing disc doming the ECM acts not as a passive constraint, but as a controllable, actively growing dynamical constraint to the tissue layer." However other studies, some of which are cited by the authors (ex: Sui et al 2018), have also shown an actively controlled degradation of local ECM as a determinant of tissue morphology. The authors should clarify in what way they contrast what they refer to as 'passive' vs 'active' roles and their results vs the already demonstrated roles of ECM.

10. In Figure 6 and Extended data Figure 8, the authors show that MMP2 knock-down results in a shift of the peripodial A-P boundary position. As a result, the regions chosen for the subsequent measurements in KD and control discs correspond to different medial-lateral positions within the disc. As can be seen in the authors' own data, cell thickness differs between the medial and lateral measurement positions even within the control discs. For the data to be more convincing, the authors should compare equivalent medial-lateral positions for all measurements in control and KD discs.

11. The authors state: "In conclusion, our results indicate that growth is uniform in the plane of the DP epithelium at different stages of larval development." This conclusion contradicts their data in Extended Fig 3d', which shows a significant difference in proliferation rates at 95-97h AEL.

Minor points:

1. The wording used to describe the changes in tissue and cell dimensions can be somewhat confusing, as the authors use both 'thickening' (at the tissue level) and 'thinning' (at the cell level) to refer to the same type of morphology change. I suggest using the words 'cell height' and 'cell width' when referring to the changes at the cell scale to clarify the authors' point.

2. The authors state: "Together, our data show a concomitant increase in tissue size with thickening and bending of the disc epithelium." This conclusion is only true for the disc proper and not for the PPE, which shows an increase in size (i.e. volume) while showing not an increase but a decrease in thickness.

3. The authors should be consistent in their use of the abbreviation DP. While it refers to the 'disc proper' in the main text, it is stated to refer to the 'disc pouch' in the SI on modelling. This is an important distinction when it comes to the authors' analysis of differential growth within or across epithelial layers, and should be corrected to prevent confusion.

4. The authors state: "The simulations indicated that increased growth in the DP centre versus the periphery can induce tissue thickening, although a 2-fold growth mismatch generates at most a 16% increase in tissue thickness (Extended 3 Data Fig.3a-b), while experimental data indicate that the DP tissue thickness doubles." It should be clarified why the authors chose to limit their assessment to 2-fold growth mismatch. If this number comes from experimental data, the data should be referenced. Otherwise the choice should be substantiated by argument.

5. In Extended Data Fig 4a, the authors provide images of the wing disc before and after the cut as evidence of its stressed state. However, as we can't infer the 'nearly instantaneous relaxation' from these snapshots, they are not very informative for making the authors' point. In addition, the provided planar view is not informative on the relaxation of the domed wing disc. If possible, the authors should provide timelapses, and section views of the discs, or should not include these images as evidence.

6. The PPE of the collagenase treated disc shown in Figure 3a and extended Figure 5a appears to be thorn. As this may potentially have a role in the disc's relaxed state, it would be better to provide another image with the PPE intact.

7. In the plot in Figure 3e (as well as some others) the y axis does not start at 0. For more accurate representation of the observed changes (or lack of them), all plots of this type should start the y-axis at 0. (In several cases this would actually help accentuate the authors' conclusions).

Reviewer #3 (Remarks to the Author):

This paper addresses the 3D morphology development, such as epithelial thickening and doming, driven by elastic deformation resulting from the differential growth between the epithelial cell layer and surrounding extracellular matrix (ECM). In particular, the authors present an interesting and novel framework for how the mechanics of ECM could affect the morphology of developing organs. They demonstrate how differential growth anisotropy in the two ECM layers of *Drosophila* wing imaginal disc can lead to thickening and doming of the wing. They show that MMP2 regulates the growth anisotropy of the ECM and thus the morphology of the developing organ. To do this, the authors use experiments and a theoretical model. The experiments seem to support their claims well. The mathematical model seems relatively simple but good enough. The model appears to match the general trends from experiments but doesn't make accurate detailed predictions. Despite this, the relatively simple model does provide some novel insights into the growth induced morphogenesis. Overall, the authors provide novel insights into how ECM mechanics can affect the morphology of developing organs. Nevertheless, I'd want authors to address the following comments and answer my questions before recommending publication.

<Major>

1. What is the rationale behind PPE schematic in Fig. S2h? Volumetric increase shown in Fig. S2g doesn't back up the flattening of PPE over time.

2. In the simulation part of Fig. S3a-b, the growth rate of pouch was assumed to be higher than hinge up to twice, but according to the continued experimental results, hinge cells feature slightly better divisions and proliferation (Fig. S3c'-d'), and finally growth rate (Fig. S3g). If experimental results are the case, what's the implication of the simulation part?

3. In Fig S5a, treatment of solely Collagenase reverses the curvature, whereas the combination of Y27632 and Collagenase does not. Does it mean inhibition of MyoII rather reinforces the tissue?

4. In Fig. S6c and Fig. S6e, how ECM volume can be approximated as discussed in the caption? That could have caused a mismatch in Fig. S6g. Here, Fig. S1g may back up this approximation, but this comparison was done for DP, not ECM. Also, will this case still hold between integrated intensity_PPE and V_PPE?

5. In Fig. 6e-d, it doesn't really make sense how experimental evidence after MMP2 KD, such as posterior shift of A/P_PPE, increased epithelial thickness of posterior PPE cells, and reduced apical cell surface area, can support the claim that MMP2 inhibition has to do with non-planar (3D) growth. Please elaborate on this.

6. Why was the ECM considered an incompressible neo-Hookean material? How is this justified given the viscoelastic and fibrous nature of ECMs?

7. In Fig. 5, the model doesn't appear to match the experimental measured thickness values. Is there an explanation on why the model differs and why this doesn't affect the conclusions?

<Minor>

1. In Fig. 1b, the orientation of anterior (left) is not consistent with what is explained in its caption (right).

2. In line 21 of page 5, it says "the model predicted that the PPE layer grew ~70% more in volume than the DP layer," but Fig. 2b notes growth rate ratio of 1.3, meaning 30%. Please clarify.

3. What was the choice for γ_{ECM} and γ_{DP} in Eqs. 2-3?

Response to REVIEWER COMMENTS

We would like to thank the referees for the time they have taken to go through the manuscript, their feedback and their helpful comments.

Here are our point-by-point answers to the reviewer's comments:

Reviewer #1 (Remarks to the Author):

"p" refers to page number and "L" refers to line number

*** General comments**

This study investigates the morphogenesis of the drosophila wing imaginal disc which undergoes both volumetric growth and bending during its development.

While the most natural hypotheses to explain the doming of growing quasi-planar tissues are either planar differential or planar anisotropic growth, several experiments on the growth rates of the different cell layers composing the imaginal disc are presented and used to motivate the need for a multilayered model based on morpho-elasticity.

The theoretical model is correct. The word "predict" occurs 8 times and it seems an overclaim. In fact, the results of the model correctly "describe" the experiment. The model shows that a description with three layers which grow differently reproduces the experimental observations, while the simpler versions tested by the authors fail to reproduce the experiments.

- We thank the reviewer for this opportunity to sharpen the language of the manuscript. In the section "*Bending is not due to differential growth of the tissue layers*" the model does not make predictions prior to any observations, so we have changed "predict" with "account for" in several instances. We feel that 'account for' clearly indicates the capability of our model to 'explain' experimental observations. Prediction is a bit too phenomenological and hence "model-free" for it to be used in this instance. Note however that in section "*Differential growth anisotropy recapitulates tissue shape changes*" the model actually provides predictions of the growth parameter ρ and of the ratio of shear moduli of ECM and DP, which are otherwise experimentally inaccessible (see Fig. 5B). In this section the word "predict" seems instead appropriate to us, and we have kept it. So overall, the current usage of the words "predict" and "account for" is more appropriate.

We take the liberty of grouping several points of the reviewer related to the question if spatial-nonhomogeneity and/or polar anisotropy of the growth tensor can explain the wing disc doming.

The authors should explicitly indicate in the introduction that they do not look for parsimony and that the complexity of their model ingredients is justified by the agreement with observations. In fact, one can imagine other simple models which could have been tested. For example a single

ECM layer growing differentially : space-dependent/anisotropic planar growth combined with an appropriate homogeneous component of the growth tensor in the Z direction. Or maybe a two layers system incorporating similar features. Since neither data on the spatial differences (in the plane) of the growth rates of the ECM, nor any data on the degree of planar anisotropy (i.e differences between radial and orthoradial growth components) of any layer are presented, such simpler models unfortunately cannot be ruled out by this study in its present form.

[...]

Although planar differential growth (spatial heterogeneity, see Klein et al, Science 315:5815 (2007)) can lead to a pronounced doming of a single layer, it should be noted that planar growth anisotropic in polar coordinates can also lead to a pronounced "doming" of a single growing layer (or for two layers growing at the same rate), see Dervaux and Ben Amar, PRL 101, 068101 (2008). This occurs for example in the algae acetabularia acetabulum where the degree of anisotropy controls the tissue curvature. Could the authors measure this anisotropy in their datasets or at least discuss this point ? Anisotropy is defined here as a difference between the radial and orthoradial components of the growth tensor.

- For the tissue layer, (to a good approximation) we can rule out spatial non-uniformity and polar anisotropy of the growth tensor since if any of them were present, the tissue after ECM digestion would remain curved or clearly show some bulging. After ECM digestion (Collagenase treatment in Fig.3b and also, in this revision, MMP2 overexpression Supplementary Fig.6), the tissue becomes nearly flat and returns to the thickness of approx. 22 μ m no matter at what time point ECM digestion is performed (Fig.3d), which is strong evidence of a return to a nearly stress-free state. The ECM digestion experiments thus show that the tissue layer is very likely a flat disc of uniform thickness, which is its stress-free state. Assuming that the relaxed state of the tissue is completely stress-free, the growth tensor is compatible (i.e. its Riemann curvature tensor vanishes identically), so there cannot be any non-uniformity and/or polar anisotropy of the growth tensor. We note that we showed previously (LeGoff, Rouault, and Lecuit 2013) that cell divisions in the wing disc exhibit a preferential orientation in tangential direction especially closely to the periphery of the wing disc. However, a quantitative translation of the cell division data of the mentioned article into a growth tensor is very challenging. Furthermore, even if the cell division orientation data were to translate into some polar growth anisotropy in \mathbb{G} , its effect on tissue shape would likely be negligible in the light of the present collagenase experiments: Since no thickness gradients or bending were observed in the tissue after collagenase, we conclude that any polar growth anisotropy in the tissue layer is not strong enough to cause significant tissue bending or thickening.
- Having ruled out polar growth anisotropy and spatial non-uniformity for the tissue, it remains to assess what happens in the ECM layer. Before discussing this aspect in detail, note that the cellular tissue has a thickness of 43 μ m, whereas the ECM layer has a thickness of 4.4 μ m, so that the tissue is almost 10 times thicker than the ECM layer. As noted by the Reviewer, is certainly true that in-plane incompatible growth (as the one originating from anisotropic and/or non-uniform growth) can produce doming. Above all if a single layer growing this way is slender, the storage of in-plane incompatibility is rapidly

relaxed into out-of-plane bending, leading for example to doming, as observed in many biological systems.

In our case, however, the tissue layer is ~10 times thicker than the ECM; the tissue certainly grows uniformly (see Supplementary Fig. 3); following growth, the whole system bends with a concavity towards the ECM at the base of the Disc Proper (DP), clearly revealing that the ECM is under tension, and henceforth showing that the ECM grows more slowly than the tissue along the planar direction. All these considerations seem to indicate that incompatibility in this system is mainly due to incompatible growth between the two layers, rather than to incompatible planar growth inside the thin ECM layer.

Also note that for the incompatible planar growth inside the thin ECM to be so vigorous as to bend the whole tissue (which is 10 times thicker), this would imply the existence of a remarkable, and highly non-trivial, growth pattern inside the ECM. Such a strong planar inhomogeneity/anisotropy in the growth of the ECM would be immediately revealed by a rather convoluted / corrugated configuration of the relaxed ECM past dissolution of the tissue (TritonX), but this is not the case, since in its relaxed configuration the ECM appears rather floppy and tensionless (see Fig.4b and Figure Rebuttal 1 below).

Overall, these considerations seem to indicate that although growth inhomogeneity / planar anisotropy of the ECM cannot be ruled out a priori, the model where the layers grow uniformly and isotropically in the plane (meaning the radial and orthoradial components of the growth tensor are identical), but differently in magnitude, is compatible with all our experimental observations.

We have also included a section in the Discussion about this (pg.12 line 19-31:

`In the present manuscript, we considered all layers to exhibit polar isotropy of the growth tensor, meaning the radial and orthoradial components of the growth tensor are identical. We note that we showed previously³⁷ that cell divisions in the wing disc exhibit a preferential orientation, suggesting some presence of polar anisotropy. However, its effect on tissue shape is likely negligible in the light of the presented collagenase experiments: Since no thickness gradients or bending were observed in the tissue after ECM digestion via collagenase, we conclude that any polar growth anisotropy in the tissue layer is not strong enough to cause significant tissue bending or thickening. Similarly, in the light of the decellularization experiments, the relaxed configuration of the ECM layer is floppy and tensionless, showing that a strong enough pattern of polar growth anisotropy to alter the shape of the ten times thicker tissue layer is absent. We conclude that any polar growth anisotropy in the tissue layer and ECM layer is not strong enough to cause significant tissue bending or thickening, which is why we assumed polar isotropy of the growth tensor throughout this manuscript.`

Example discs after Decellularization (TritonX)

Figure Rebuttal 1 – The ECM does not form any higher structure upon decellularization

Optical cross-section of three 118hAEL example discs after 15min Decellularization with Triton-X100. The Remaining cellular tissue is marked in magenta (anti-Wg/Ptc staining) and the ECM is marked by Collagen IV (Vkg::GFP, green). Sections of the bottom ECM_{DP} that have lost connection to the DP tissue layer are marked by red bars. Remark that the relaxed bottom ECM seems sloppy and does not form any higher structure.

* Main text

Experiments are careful. This part of the text is clear and written with many details.

There should be one short sentence, aimed at non-specialists, to summarize the exact mechanism for bending, according to the authors. Or make an analogy with daily life objects.

- We thank the reviewer for this comment. In the last paragraph of the introduction, we added a few lines of simpler explanation, and drew an analogy with a bimetal thermostat (pg.3 line 7-9):

`This is comparable to the functioning of a bimetal thermostats, where the differential thermal expansion properties of two connected metal layers lead to bending.`

If (x,y) is the plane of the tissue in cartesian coordinates, (r,θ) the same plane in polar coordinates, and z the perpendicular direction, "anisotropic growth" can either mean :

- growth along x is different from along y : elongation, or extension compression
 - growth along r is different from along θ : polar anisotropic growth
 - growth along Z is different from along x,y : in-plane growth differs from out-of-plane thickening
- In the latter case, which is extensively studied in this manuscript, it might be better to avoid the words "anisotropy" or "anisotropic growth" which are an unnecessary source of confusion.

- This is a very helpful comment and we feel it adds clarity to explicitly state that it is not polar anisotropy we are considering. We have added this in the paragraph just after main text Eq. (1) (pg.5 line 23-25), which we feel is now very clear about the type of growth anisotropy modelled here. We feel that it is justified to stick with the term "growth anisotropy" which is used in the plant (Hervieux et al. 2016) and tissue mechanics (Ben Amar and Jia 2013) community:

`Notice that we do not consider polar growth anisotropy, in which case the radial and hoop components of the growth tensor, i.e. the first two entries of the diagonal matrices in Eq. (1), would have been different one from the other.`

- p2 L11 :

It would be useful to mention here the effect of differential growth in plants (including flowers, see e.g. the papers of Enrico Coen).

Some parts of the Discussion (e.g. p11 L25) could be moved to Introduction, where differential growth in general is presented.

The Discussion should rather focus on the perspectives opened by the current model, which are much more specific.

- We have added two examples in which differential growth in a tissue layer coupled with mechanics creates forms in plants (including Coen's work, pg.2 line 11-15):
`Differential growth refers to the situation where some parts of a tissue grow faster than others, building up mechanical stress and changing the shape of the tissue. Examples of differential growth in the plant world are the formation of flower petals⁹ and leaves¹⁰. In the animal world, differential growth can be seen, for example, in the formation of solid tumours¹¹.`
- Furthermore, we have removed from the discussion statements that echo ideas already present in the introduction, namely the passage p11 L25 referred to by the reviewer, as well as examples of differential growth between epithelial layers in the 2nd paragraph of the discussion.

- section "Tissue thickening is not due to differential cell growth in the tissue plane" :

In this section, the hypothesis that planar differential growth drives tissue thickening is investigated. It is shown that, using a crude space-dependent growth tensor consisting of an inner fast growing core surrounded by an annulus growing at a slower pace), the observed thickening cannot be reproduced with reasonable growth rates differences between inner and outer sections. While this is interesting, the most natural hypothesis for the thickening of the tissue is simply a homogeneous growth process in the "Z" direction. Although this natural hypothesis is mentioned, and ruled out, later (in the "The ECM is essential for tissue bending" section) it would be better to discuss this hypothesis right at the beginning of this section.

- We thank the reviewer for this comment. We have expanded the first paragraph of the mentioned section in two ways: Firstly, we have added a sentence right at the beginning of this section mentioning growth in the "Z"-direction as a possible explanation of the thickening. Secondly, we motivated our choice to consider first the non-uniform growth of a tissue layer based on this being the primary explanation of buildup of stress in the wing disc in previous literature in the final sentence of the same paragraph (pg.4 line 2-10):

`We first investigate what drives the doubling in DP thickness. Tissue thickening of the DP could theoretically result from two scenarios: either cells actively increase their height via z-growth or the observed increase in cell height is a result of elastic compression due to cell crowding. Previous work proposed that cell proliferation is increased in the centre versus the periphery of the disc, leading to crowding and stress accumulation in the disc centre³⁴. Consistently, artificially increasing growth in clones of cells leads to the accumulation of stress in neighbouring cells though not to an increase in thickness^{35,36}. We therefore first considered that the doubling in tissue thickness stems from cellular compression due to spatially non-uniform growth in the plane of the DP tissue.`

- section "Discussion"

Perspectives should be discussed more precisely. In principle, could the present study be extended to describe the fact that the wing disc is eventually fully folded (p10 L13 or p10 L29, explain what additional ingredients would be required) ?

- We have reworked the discussion and now discuss in more detail the limitations of the current model (pg.12 line 1-18) and to which extent it could be modified in order to address related biological questions, in particular disc evagination (pg.12 line 32-37):

`We show that a nearly incompressible neo-Hookean model, in which pre-stress is added through differential growth anisotropy, can well recapitulate 3rd instar wing disc development. The assumption of incompressibility was based on measurements of ECM incompressibility in decellularization experiments (Supplementary Fig. 9c"). We use the neo-Hookean model as it is the simplest model of non-linear (finite strain) elasticity. But indeed, some ECMs can display strain-stiffening properties⁵⁵. Generalized neo-Hookean models that capture this effect (Fung and Gent models) introduce a strain-hardening parameter which requires careful extension tests of the bottom ECM that were beyond the scope of this study.`

We aimed to keep the number of model parameters small and made several simplifying assumptions. Firstly, we decided not to model the hinge region as it grows at similar rates as the wing pouch (Supplementary Fig.3). We also assumed that the wing disc is axisymmetric and has a no-traction (i.e. force-free) boundary condition at the disc edge. This simplifying assumption is more correct in Anterior-Posterior direction than in Dorsal-Ventral direction, where complex folds form in larger discs⁵⁶. Capturing the whole morphology of the disc will be an interesting challenge for future studies. Nevertheless, our modelling choices, guided by relative simplicity, were sufficient to understand key physical experiments like collagenase and decellularization results and to capture quantitatively key geometric quantities and predict the degree of growth anisotropy and stiffness ratio between the layers.

[...]

More complex versions of our model could provide insight in related biological questions such as the 3D shape remodelling process of disc eversion. Actomyosin contractility will be important to consider as supracellular cables in the periphery of the doming wing pouch³⁵ were proposed to form a constrictive ring required for disc evagination. It is possible that tissue bending induced by the ECM may induce actomyosin supracellular cables due to tissue stress accumulation and mechanosensitivity of actomyosin networks.

*** Figures**

- Fig 1b, p16 L8 :

recall here the definition of hAEL (currently it is only defined at p3 L7).

- We have added the definition of hAEL in the figure caption.

*** Methods**

- p26 L10 : grammar

*** Extended Data Figures**

- p43-44 : Fig 6d should be in chronological order, unless there is a good reason for antichronological one.

- Grammar has been fixed and reordering has been done.

*** Supplementary information / Elastic model :**

The authors should re-read it to bring it to the level and clarity of the main text.

This part seems to be written much more hastily than the main text :

- inconsistent notations (both J_A and Δ) are used to denote the determinant of the elastic part of the deformation tensor),
- alternative definitions (r and θ can indifferently be called either "polar" or "cylindrical" coordinates but not both),
- unclear or allusive sentences, in particular (but not only) the paragraph "With the parameters" (p4)
- equations lacking explanations,
- several typos, spurious or missing spaces,
- low legibility figures (check font size),
- lack of transition between two paragraphs,
- calling figures in main text and in Supp Info with the same numbers, resulting in confusion,
- etc.

- We thank the reviewer for the thorough reading of the SI and for making these suggestions. We believe we have addressed them all by reworking the SI thoroughly. Notice that while we made considerable changes in an effort to better explain and motivate our equations and modelling assumptions in Section M1 (Description of the multi-layer wing disc model), we left the details of numerical implementation (Section M2) rather brief in terms of explanations. The reason is that Section M2 serves to recapitulate some known techniques and results (implementation of Growth in Comsol, analytical solution of an incompressible neo-Hookean disk growing with polar anisotropy), and the reader can find more detailed explanations in the cited references.

- section 1.2.1. of the SI Elastic model:

The conclusion at the end of this section is surprising and seems to contradict both analytical and experimental results. It was demonstrated that either anisotropic (in cylindrical coordinates) or isotropic but space-dependent growth leads to a pronounced "doming" of a single growing layer. The results shown in this section, which lead to a different conclusion, might be a consequence of the peculiar growth tensor (consisting of an inner fast growing core surrounded by an annulus growing at a slower pace) chosen here. But this is not a general conclusion and cannot motivate the need for a more sophisticated model.

Surely the authors do not mean that doming cannot be reproduced with a space-dependent tensor (which is likely incorrect) but rather that the measured thickening of the system cannot be reproduced with a simple space-dependent planar growth tensor.

In the same section, it is also noted that the softness of the material excludes a doming. This is surprising since, in the limit of an incompressible material made of a single layer, the final shape of the growing layer should be independent of the stiffness of the layer (the stiffness is only a prefactor which fixes the magnitude of the stresses). Can you also clarify this point ?

- We would like to clarify that in this part of the manuscript our prior intention was to focus on tissue thickening due to elastic compression induced by non-homogeneous growth and not on doming. The use of the single layer simulation (previously shown in Supplementary Fig.3b) was exclusively to evaluate the needed difference in the spatial magnitude of growth to induce a 2-fold tissue thickening. As these simple simulations have demonstrated, even a two-fold difference in growth magnitude would only lead to a marginal increase of ~16% in central tissue thickness.

However, the reviewer is correct in questioning the premise of the simulations shown previously in Supplementary Fig.3 a+b and discussed previously in the SI section 1.2.1. Indeed, neither are they general enough to make a statement about the possibility of doming, nor do they apply to the present specific case in which spatial non-uniformity in the DP has been clearly experimentally ruled out (Supplementary Fig. 3). We therefore remove these simulations, which are no longer present in Supplementary Fig.3 as well as the SI model Section M1.2.

If the reviewer, and the editor, think that based on our argumentation above it would nevertheless be beneficial for the reader to see these simulations, we can include them again.

Stress free situations can have different causes. The legend of Fig 2b states that the grown system is stress free if the different layers grow at the same rate. But this is obviously not true, in all generality, if the growth tensor is either anisotropic (in the plane) or space-dependent. The caption does not specify whether the growth is heterogeneous or homogeneous; on the picture the growth is apparently heterogeneous. Can you also clarify this point ?

- The legend of Fig.2b implicitly assumes a growth tensor of the form Eq. (1), which exhibits neither polar anisotropy nor spatial inhomogeneity. The reference to Eq. 1 is now explicitly made in the legend of Fig.2b.

- The sentences before and after Eq. (33) should be reconsidered. It should be mentioned that Ref. [5] is an anterior work of one of the authors. Hence, if the authors have noticed a mistake, they should first publish an erratum (the procedure is simple and quick), and then quote here both Ref. [5] and its erratum. This is a correct scientific practice.

- We have sent a request for an erratum to the Bulletin of Mathematical Biology (where the article in question was published), where it is currently in the hands of Shenbagam Selvaraj who is responsible for post-publication corrections at this journal. If Springer decides to publish the erratum we will add a reference to it in the Supplementary Information of the present article.

Reviewer #2 (Remarks to the Author):

Harmansa et al, investigate the role of differential growth within epithelia and across the epithelial and extracellular matrix layers in determining the final domed morphology of the wing imaginal disc. Combining a large experimental dataset with quantitative measurements and computational modeling, they propose that the domed shape of the wing pouch can't be explained by differential growth rates within the tissue layer, but by differential growth between the epithelia and the basal extracellular matrix. The role of local ECM regulation in wing disc growth and morphogenesis is already demonstrated by others. Therefore, the data need to convincingly establish that ECM growth anisotropy is actively controlling the final tissue shape. Several of the model assumptions are not very well supported and additional experimental data are needed to back up the authors' conclusions. While the authors' work provides new information to advance our understanding of the role of different tissue layers in epithelial morphogenesis, these issues must be addressed for the work to be recommended for publication.

Major points:

1. The authors model the growth of the tissue layers as in plane only. While it is well established that cell proliferation is indeed in-plane, this does not imply that growth take place in the plane. In addition, the authors' own data clearly show that there is substantial non-planar tissue growth (i.e. increase in thickness). It is not at all clear why the authors think the observed DP growth in z-axis is dispensable for their model. How would the model predictions change if the authors indeed taken DP growth in z into consideration?

- Indeed, disc thickness increases during development (Fig.1c). However, whether this increase is due to growth in z-direction or to a mechanical, elastic response of the tissue (e.g. due to compression) cannot be decided alone from observing wild type disc morphology. This is because the observed morphology is a result of growth and elastic deformation due to mechanical stress.

In order to visualise the specific contribution of growth to shape, one needs to eliminate the mechanical deformation by observing the non-stressed, relaxed state of the structure. Acute digestion of the ECM layer by Collagenase (Fig.3) clearly demonstrates that the DP epithelium does not actively grow in thickness since ECM digestion at different developmental stages consistently results in relaxation of epithelial thickness to a value close to the value observed at 65hAEL (~22 μ m, see Fig.3d). Therefore, growth drives expansion in the x/y-plane only, while the observed thickness increase is due to elastic deformation of the cells by the constraining ECM layer.

One may note however that at 96hAEL following ECM digestion, the relaxed thickness does not fully drop from ~43 μ m to 22 μ m but to ~26 μ m (Fig.3d, right). This could be potentially interpreted as minor thickness growth of the DP epithelium. However, the experiment shown in Fig.3 is an acute digestion of the ECM and the observed disc shapes are acute relaxation (only 2 minutes between ECM digestion and tissue fixation). Therefore, it is indeed important to further investigate if this thickness increase is a result

of minor thickness growth or due to slower relaxation dynamics (e.g. due to a stabilising cytoskeleton).

In order to address this point we have genetically removed the ECM layer to allow for longer relaxation times. We temporally controlled overexpression of MMP2 in the posterior compartment using a *Gal80ts* transgene to induce MMP2 overexpression 12 hours before dissection and fixation. We have quantified disc thickness upon MMP2 overexpression for 12 hours at 118hAEL and find that the relaxed thickness in the posterior compartment is reduced to values $\sim 22\mu\text{m}$ similar to 65hAEL (see New Supplementary Data Fig.6). This finding strongly supports the notion that epithelial growth in the DP is planar with no active growth along the z-axis.

Furthermore, the authors use experimental data on volume change rates to determine the γ_{DP} , while presuming $\gamma_{\text{Z}} = 1$ in their calculations. Knowing that thickness increase is a major contributor to the volume changes measured experimentally, this would significantly distort their estimation of γ_{DP} used in their model. Using measurements of area change over time would be a better estimate for γ_{DP} .

- We assume that we have not been clear in the text on how γ_{DP} has been estimated and what it describes. γ_{DP} describes the stressless volume expansion of the DP tissue in the given direction. Volume changes have been carefully estimated on the DP tissue level in 3D (see Supplementary Data Fig.1a-g).

In the previous point we have summarised the multiple evidence establishing that the DP does not significantly grow in thickness and hence that $\gamma_{\text{DP}}^{\text{Z}}$ (describing the stressless growth in thickness) equals 1. Therefore, volume expansion due to cellular growth and proliferation results in stressless tissue expansion in the x/y-plane. In the model γ_{DP} therefore describes the experimentally measured volume expansion of a tissue layer growing without any external constraint.

Again based on experimental data on volumetric growth, the authors use a γ_{ECM} that is lower than the γ_{DP} in their model. However, the area used in DP and ECM volume calculations is apparently the same throughout the study. Wouldn't this imply similar rates of planar growth for DP and ECM?

- Volumetric growth cannot be inferred from area because the thickness responds to the mechanical stress exerted on the layers. So we need to distinguish measurements in the stress and relaxed configurations.
- Importantly, the observed area in the *stressed configuration* (e.g. at 96hAEL) are identical for the DP tissue and the bottom ECM since they are physically linked via integrins. Furthermore, the DP tissue is under compression while the bottom ECM is under tension. However, the *relaxed area* differs for the tissue and the ECM layer. As we have shown in Fig.3, the DP tissue expands when separated from the ECM layer. In contrast, the ECM layer contracts when separated from the DP layer (see Fig.4d). Therefore, growth of the DP and its ECM layer is not 'compatible' (meaning not stress-free) and therefore, the planar growth rates for the DP and the bottom ECM must differ.

Another, maybe more intuitive way is to compare stressed and relaxed thicknesses of both layers. While the DP tissue layer consistently decreases in thickness upon ECM removal (Fig.3), ECM thickness significantly increases upon tissue removal (Supplementary Fig.9b). Given these inverse trends, these two layers must experience different kinds of stresses: compressive for the DP tissue and tensile for the bottom ECM.

It appears that estimating planar growth rates based on volumetric growth data is inflating the difference between the DP and ECM planar growth rates used in the final model. Could this inflated planar growth difference itself, without the anisotropic growth difference, explain the doming effect? (i.e. would the scenario given in SI 1.2.4 predict the final tissue shape if the γ_{ECM} value used was not higher but same as the one used in SI 1.2.3?)

- First, we would like to clarify that the magnitude of ECM growth is identical in simulations in SI 1.2.3 and 1.2.4. Simulations only differ by the degree of ECM growth anisotropy (planar for 1.2.4 and non-planar for 1.2.3). The scenario where growth of both layers is planar (see SI 1.2.4) but the magnitude of growth differs by 20% is shown in Supplementary Figure 8e. Such simulations lead to disc morphologies that are not sufficiently bent and epithelial height is too small. Therefore, the experimentally measured difference in growth magnitude of 15-20% (Supplementary Figure 7d and 8d) is not sufficient to explain disc shape via planar ECM growth.

Overall, the choice of taking $\gamma_{ECM} \neq 1$ while keeping $\gamma_{DP} = 1$ is a key assumption of the model that is the bases of most of the authors final conclusions. However, this choice does not appear to be substantiated in the experimental data. This issue must be experimentally resolved.

- We appreciate the critical input of the reviewer and we have added further experimental evidence showing that indeed, the DP tissue follows planar growth. We show that genetically induced ECM digestion via MMP2 overexpression for 12 hours leads to a relaxation in tissue thickness close to the values observed at 65hAEL (~22 μ m) even in fully grown discs (118hAEL). This new experimental data is now included in Supplementary Figure 6. We believe that this solidifies our main conclusion and highlights the crucial aspect of growth anisotropy in epithelial morphogenesis.

2. The authors use cultured wing discs to show that clonal growth rates are consistent along the disc proper, and across disc layers. However, they do not establish that cultured discs go through the same morphological changes as discs *in vivo*. Indeed, for the DP clones that are shown in Figure 2c and Extended Figure 3 e, we do not observe the thickness changes reported for the disc proper in Figure 1. To support the authors' conclusion that differential growth is not the cause of thickness increase and the domed shape of the disc, it should be shown that the discs can indeed go through thickening and doming *ex vivo*.

- We appreciate this comment and have investigated disc bending and have quantified DP clone thickness in our *ex vivo* movies.

Indeed, we observe that *ex vivo* cultured 72hAEL discs show increased bending over the 10h culture period (see Figure Rebuttal 2). Furthermore, we have quantified clone thickness at the beginning and after 10h of culture and find that clone thickness increases. The observed thickness increase is slightly reduced in *ex vivo* culture compared to *in vivo* development ($30.4 \pm 2.3 \mu\text{m}$ *in vivo* at 80hAEL compared to $27.4 \pm 3.0 \mu\text{m}$ after 10h of culture). Therefore, discs consistently undergo thickening and doming in *ex vivo* culture.

Figure Rebuttal 2 - *ex vivo* bending and height increase

(a) Representative wing disc containing clones marked by cytosolic GFP (GFP_{cyto}) and Histone::RFP (Hist::RFP), dissected at 72hAEL and cultured *ex vivo* for indicated hours in plane (top) and section view (bottom). The basal outline is marked by a red dashed line in the section view. Explants of 72hAEL old larvae are relatively flat at beginning of culture (left) and bend during the culture period. (b) Quantification of clone height at beginning of culture (0h) and after 10h of *ex vivo* culture (10h) of a disc dissected at 72hAEL. Cell height significantly increases during the 10h culture period. Only two clones did not show increased height (red lines).

In addition, the timing chosen for the *ex vivo* experiments does not correspond to the time of maximal change in thickness or doming behaviour. To strengthen their point, the authors should show clonal growth data from later time points (i.e. 96-118h).

- Quantifying clonal growth in *ex vivo* culture is very time consuming since 3D volume segmentation of a movie in Ilastik requires quite some manual correction. We therefore have focused on the 72hAEL time point since this is the moment when disc thickness starts to increase and bending is initiated. We see that further quantification would be interesting to provide a more detailed view. However, genetically reducing PPE growth (Supplementary Fig.4) does not abolish disc bending and therefore rules out a dominant role of the PPE in disc bending (please see also response to point 3.).

Indeed, the results shown for growth rates in DP and PPE at 72h in Figure 2c' and authors conclusions on them, apparently contradict the authors' data shown in Extended Figure 2f and g, where the authors report a clear cellular volume increase for DP from 72 to 96h while the PPE volume remains unchanged. How do the authors reconcile this contradiction? The lack of PPE cell volume change from 72h to 96h AEL (Extended figure 2g) is also in apparent disagreement of the continuous PPE volume increase from 72-118h AEL at the tissue scale

(Extended Figure 1j). The authors should comment on their interpretation of these differences and justify their choice of using the clonal growth rates measured *ex vivo* at 72h to verify their model.

- This is a very careful observation that cell volume increases in the DP while it remains constant in the PPE between 72 and 96hAEL. However, it is important to keep in mind that growth rates cannot directly be related to cellular volume since average cellular volume depends on cell division rates. We have quantified that cell cycle length continuously decreases in the DP (Supplementary Fig.3c+d), consistent with a slight increase in average cell volume (former Supplementary Fig.2f). In contrast, in the PPE the situation is more complex since PPE cells were suggested to stop dividing and become polyploid, which explains the jump in PPE average cell volume between 96 and 118hAEL). Therefore, it is not trivial to connect average cell volume to growth data without precisely knowing division times. Such investigations exceed the scope of this study and will require careful, quantitative characterization of growth and division dynamics observing large sample numbers.

Given the mentioned points and that this data is not required for the understanding of the manuscript, we have decided to remove this data from the MS in order to avoid confusion.

Could the authors use the differences in volume change rate observed at the tissue scale over the time of disc development to verify their model? Considering the limitations of the *ex vivo* system used to measure clonal growth rates, the tissue scale measurements done *in vivo* would be more accurate and more relevant.

- We want to clarify that *ex vivo* clonal growth rates have exclusively been used to test model predictions (i.e. test planar growth differences or growth differences between the tissue layers).

However, we have used *in vivo* volumetric measurements on the tissue scale (see Fig.1d and Supplementary Fig.1a-g) to estimate the γ_{DP} parameter of the growth tensor.

3. The authors block proliferation in PPE to show that a higher growth rate of the PPE is not the cause of disc doming. However, by doing so, they presumably create a growth differential in the other direction (i.e. the DP now has a higher growth rate than the PPE). According to the authors' model, wouldn't this differential result in bending of the disc in the other direction? One explanation is that, in response to growth inhibition in the PPE, the growth of DP is altered as well (as is suggested by the apparently smaller overall size of the disc shown in Extended Figure 4d). The authors should check if this is the case, and demonstrate that this does not impact the interpretation of their results.

We indeed presented a rather misleading image that suggests that DP growth is reduced upon growth inhibition in the PPE. We have repeated this experiment and have now quantified the DP and PPE area. Indeed, while the average PPE area significantly decreases (Supplementary Fig.4e), DP area remains unchanged when reducing growth in the PPE (Supplementary Fig.4d).

Furthermore, we have now quantified and compared the relative volume increase between the PPE and DP tissue to further investigate a possible growth differential. Indeed, we find that also in control discs the PPE tissue overlying the DP grows significantly less than the underlying DP tissue (see Supplementary Fig.1k). Hence, as hinted by the reviewer, there is indeed a growth differential between the PPE and the DP tissue. This growth differential possibly explains the decrease in PPE thickness during disc growth (see Fig.1c') due to a stretching of the PPE layer during disc bending. Importantly, this growth differential does not significantly impact disc mechanics and morphology since upon ECM digestion and MyoII inhibition the wing discs flatten (and do not form an inverse dome, see Fig.3b). Therefore, despite a growth differential, the much thicker DP layer dominates over the much thinner PPE layer. These observations clearly demonstrate that another component (the ECM!) is required to induce symmetry breaking and bending.

We have rewritten the corresponding section in the main text (pg.5 line 34 following):

'In order to experimentally test this prediction we measured growth rates in DP and PPE clones in ex vivo cultured discs (Fig.2c). However, at the time point of bending onset (72hAEL) clonal growth rates over a period of 9 hours were comparable in both epithelial layers. We next quantified the temporal volume increase of the PPE tissue that covers the DP pouch region between 72 and 118hAEL (Supplementary Fig.1.h-j). Interestingly, we found that after 72hAEL the PPE volume increases significantly less than the DP volume (Supplementary Fig.1k), suggesting that the PPE layer possibly acts as a constraint for DP expansion. To test the mechanical role of the PPE layer, we further reduced peripodial growth by overexpression of a dominant-negative form of Phosphatidylinositol 3-kinase (PI3K_{DN}), a major growth regulator. However, reduced peripodial growth did not abolish disc bending (Supplementary Fig.4b-h). These results demonstrate that epithelial thickening and the direction of doming cannot be explained by non-uniform growth within or between epithelial layers. In particular, our finding that volume growth is reduced in the top (PPE) versus the bottom layer (DP) suggest that epithelial growth dynamics might favour inverse bending of the disc tissue. This raises the question which other component/constraint is responsible for guiding the upward doming of the wing disc.'

We also added a section in the discussion elaborating on the mechanics of the PPE tissue layer (see pg.11 line 23 following):

'We have studied the relative volume growth as well as the anisotropy of growth of both the tissue and ECM layers of the disc. We find that the two tissue layers follow planar growth, though the DP expands more in volume than the PPE leading to a growth differential between these two layers. Therefore, the PPE layer is stretched by the stronger growing DP layer providing an explanation for the decrease in PPE thickness during development (Fig.1c') However, this difference in tissue growth does not significantly impact the mechanics driving disc morphology since upon ECM digestion and MyoII inhibition disc doming is lost (Fig.3b), suggesting that the thick DP layer mechanically dominates the much thinner PPE. Strikingly, we find that also the top

ECM_{PPE} layer follows planar growth, compatible with the mechanically dominating DP tissue layer.'

4. The authors estimate PPE ECM volume by multiplying disc area by ECM thickness. However, as the authors note, the image in Extended Figure 6d shows significant wrinkling of the ECM layer at 118h AEL. This suggests an increase in ECM volume (i.e. growth) that is not taken into consideration in the growth measurements. As the similar growth rate of the PPE and DP tissues and the PPE ECM is one of the major aspects of their model and their findings, the authors should account for the ECM growth that is represented in the PPE ECM wrinkling. Could the authors use Vkg intensity to measure growth of PPE ECM as they have done for DP ECM?

- We are grateful to the reviewer to point out that indeed, a simple estimation of area*thickness does not hold true for the top ECM due to wrinkling. We have created a quantitative data set to measure growth of the top ECM_{PPE} by assessing changes in Vkg::GFP integrated intensity. This data is now included in Supplementary Fig.8a-b). Importantly, as rightly guessed by the reviewer, using Vkg::GFP intensity measurements the top ECM shows increased growth compared to the PPE tissue layer. We find that the top ECM intensity increases ~2.5-fold more than the PPE volume, likely explaining the wrinkling towards the end of development (as seen in Supplementary Fig.7b). In contrast, we find that the peripodial integrated Vkg::GFP_{PPE} intensity increases to a very similar extent as the DP volume (see Supplementary Fig.8b, magenta data points, linear fit with slope of 0.95).

This last observation indicates that the top ECM_{PPE} does not act as a geometric constraint for bending since the magnitude and the anisotropy is very similar to the DP layer (and hence growth between these two layers is compatible).

This new finding is now described in the section 'Differential volume growth between DP and associated ECM' on pg.7 line 19-30:

'Next, we quantified the growth of the bottom ECM_{DP}, focusing on the wing pouch (marked by the ring of Wingless expression), following two approaches. First, we estimated the ECM_{DP} volume by multiplying the ECM thickness with the pouch area (Supplementary Fig.7d, see methods for details). Secondly, we measured the increase in integrated fluorescence intensity of the Vkg::GFP signal underlying the wing pouch (Supplementary Fig.8c). Comparing the relative increase in DP volume with the increase in ECM_{DP} estimated volume or integrated intensity showed that the DP layer outgrew the ECM_{DP} by ~15-20% (Supplementary Fig.7d and 8d). In contrast, when quantifying the changes in integrated Vkg::GFP intensity in the top ECM_{PPE} we observed that the top ECM_{PPE} intensity increases ~2.5-times faster than the PPE tissue volume (Supplementary Fig.8a-b and see discussion). Notably, we observed a very similar increase in top ECM_{PPE} intensity and DP volume (Supplementary Fig.8b), suggesting that the top ECM layer grows compatible with the DP tissue layer.'

5. The authors conclude that the difference in growth anisotropy is key in explaining the domed morphology of the disc. The increase in ECM thickness would suggest an increase in stiffness as well. Could the authors exclude a role of ECM stiffness?

- The role of ECM stiffness compared to DP stiffness is absolutely crucial and is studied in detail in Fig. 5B, which is a phase diagram with anisotropy represented in the horizontal axis and the ratio of ECM stiffness divided by DP stiffness, μ_{ECM}/μ_{DP} , on the vertical axis. Through a procedure outlined in detail in SI Section M1.2.2, we determined as a best fit a stiffness ratio of $\mu_{ECM}/\mu_{DP} = 25$, meaning that the ECM is 25x stiffer than the DP.

6. The authors favor a role of ECM over contractility in the control of bending based on acute inhibition of contractility and ECM degradation. Yet these results could be interpreted as a function of ECM in the maintenance of DP shape. The authors need to characterize the apical and basal actomyosin distributions during DP growth and analyze the impact of MyoII inhibition on clone and tissue growth. A detailed description of the previous findings on the role of ECM and actomyosin contractility (Nematbakhsh et al, 2020) need to be provided.

- Unfortunately, there is no way to interfere with Myosin II contractility without blocking cell division (and hence growth, since MyoII is also required for cytokinesis). Therefore, it is not possible to evaluate the requirement for MyoII on a longer time scale. Importantly, also the study mentioned by the reviewer (Nematbakhsh et al. 2020) lacks compelling evidence that MyoII contractility is required for disc shape in the long-term. Indeed in a more recent study the authors provide evidence that local tissue bending *correlates* with increased basal MyoII levels ((Kumar et al. 2022) Fig.4d and Fig.5). However, these results remain a pure correlation and cannot be used to evaluate the requirement for MyoII for local bending. Importantly, the study by Nematbakhsh *et.al.* did not consider epithelial and ECM growth in their model, leaving basal MyoII contractility as the only possible hypothesis to explain tissue bending. Besides this, we show that inhibition of MyoII does not significantly alter the global bending of the wing disc while ECM digestion clearly does. Also, digestion of the ECM alone, leaving MyoII intact results in an inversion of disc curvature (Supplementary Fig.5a-b). This clearly indicates that basal MyoII tension does not actively deform the disc and increase local disc curvature. Collectively, these observations strongly support a dominant role for tissue growth and the ECM in determining disc shape.

7. The DP bending and the DP shape are not quantitatively compared with the ones obtained by modelling (Fig. 5 and Extended Data Fig 6h). Quantitative comparisons of the DP bending in vivo and in silico in the different experimental conditions is needed to validate the authors' conclusions.

Our model produces quantitative results for the DP thickness, ECM thickness and ECM thickness increase upon decellularization (Fig. 5c). What these quantities have in common is that they are measured at the axis of symmetry of our modelled wing disc. However, to

measure the disc bending, we must rely also on information closer to the edge of the wing disc. Here, we made a number of simplifying assumptions: we excluded the surrounding hinge tissue from modelling and furthermore assumed axisymmetry of the wing disc, which in reality has a rather ellipsoidal form. In the light of these simplifications, precise local curvature cannot be captured quantitatively by our model on the entire wing disc domain. That said, with the simplified geometric assumptions we have put into the model, we not only obtain good quantitative results for the aforementioned quantities (DP thickness, ECM thickness, ECM thickness increase) but more importantly, we are capable of understanding physically important experiments such as collagenase digestion and decellularization. Furthermore, this model allows us to understand DP thickening as a passive compression phenomenon rather than an active thickness growth of cells. We therefore feel that the simplifying assumptions are justified, even if the model cannot quantitatively replicate curvature everywhere in the wing disc.

8. The authors provide perturbation of ECM growth in the PPE will lead to tissue bending. However, they do not demonstrate that introducing ECM anisotropic growth in the DP is necessary to promote dome shape formation. The author needs to directly test the role of ECM growth anisotropy in the DP to substantiate their conclusions.

- First, it is important to recall that it is not the fact that ECM growth is anisotropic that yields tissue deformation, but it is the actual difference in growth anisotropy between the connected layers. While the DP tissue grows *planar* (maximally anisotropic), the bottom ECM_{DP} follows non-planar growth and increases in thickness (and hence grows more isotropically than the tissue layer). It is the difference in planar tissue and non-planar ECM growth that leads to geometric incompatibility, residual stress accumulation and tissue deformation.
- We have demonstrated that MMP2, which is expressed at higher levels in the PPE layer, acts as a regulator of ECM growth anisotropy in the peripodial ECM. Indeed, an experimentally induced reduction of MMP2 levels in peripodial cells results in non-planar top ECM growth, leading to alterations in epithelial geometry, namely an increase in epithelial thickness and a decrease in apical cell surface area (as outlined in Fig.6). Following the suggestion of the reviewer, we have also attempted to modulate the growth anisotropy of the bottom ECM by modulating MMP2 levels in the DP epithelium. As shown above, the ECM is very sensitive to overexpression of MMP2 (see Point 1) and overexpression of MMP2 for only 12 hours results in a complete loss of the ECM layer (see Supplementary Fig.6). Ideally, we would have aimed for a very mild overexpression of MMP2 that, similar to the peripodial layer, would yield a planar expansion of the bottom ECM. However, this has proven experimentally not feasible. Nevertheless, the fact that MMP2 overexpression leads to a loss of disc doming and thickening (Supplementary Fig.6) clearly supports the notion that ECM growth has to be tightly regulated to yield correct disc morphology. Along this line, we have knocked down MMP2 in the DP layer (using *nubbin-Gal4*) in order to test if we could decrease the anisotropy of bottom ECM growth (e.g. making the ECM grow more isotropic, see Supplementary Fig.10g-j). Importantly, MMP is only expressed at very low levels in the DP layer (see Fig.6A) and we

therefore expected a minimal impact of MMP2 on DP morphology. Still, when we reduced MMP2 levels in the DP cells by RNA-interference we observed a significant increase in ECM intensity and thickness (see Supplementary Fig.10j). This suggests that indeed MMP2 is required for tuning bottom ECM growth anisotropy and inhibition of the low MMP2 levels in the DP further increases the ECM growth in z-direction. Moreover, we observe that on average knock-down discs show more uniform and slightly increased curvature as controls (see Supplementary Fig.10i). These findings support the notion that growth anisotropy of the bottom ECM layer needs to be tightly regulated to yield correct organ morphology.

9. The authors write “These studies suggest a universal mechanism where a passive ECM layer constrains epithelial growth and controls organ morphogenesis. Here, we demonstrate that during *Drosophila* wing disc doming the ECM acts not as a passive constraint, but as a controllable, actively growing dynamical constraint to the tissue layer.” However other studies, some of which are cited by the authors (ex: Sui et al 2018), have also shown an actively controlled degradation of local ECM as a determinant of tissue morphology. The authors should clarify in what way they contrast what they refer to as ‘passive’ vs ‘active’ roles and their results vs the already demonstrated roles of ECM.

- We agree that in particular the work of the Dahmann lab (Sui et al. 2018) has highlighted the role of the ECM in wing disc folding. However, Sui *et.al.* describe how local degradation of an existing ECM can yield 3D shape changes. Indeed, this modulation (i.e. digestion) is also induced by the cell layer. Nevertheless, the EMC appears as a passive constraint that needs to be removed locally in order to allow folding. Analogously, several other previous studies have suggested a mechanism where epithelial growth constrained by an ECM can induce tissue bending (Oltean et al. 2016; Lane et al. 1993; Daley and Yamada 2013; Grobstein and Cohen 1965; Hayakawa, Kishi, and Nakanishi 1992). However, in all of these cited studies the ECM was considered as a non-growing constraint (hence ‘passive’). In contrast, in this study we highlight the ECM as a growing structure whose growth can be *actively* modulated by the epithelial layers. As we write, during wing disc growth the ECM acts as an ‘actively growing dynamic constraint’. Hence, we are referring to the fact that growth of the ECM is not passive (i.e. solely depends on deposited Collagen fibres that self-organise on the basal disc surface) but can be *actively modulated and regulated* by the cell layer. Moreover, the tissue controls the growth anisotropy of the ECM layer. Therefore, epithelia possess the capacity to tune ECM growth from either planar (which is stress-free) to non-planar 3D growth, which can lead to stress accumulation and 3D shape changes.

We have outlined this in a few sentences in the discussion now (pg.11 line 17-22):

“Previous studies have highlighted a role of local ECM degradation in the formation of epithelial folds or that ECM layers can act as constraint for epithelial tissue expansion leading to cell and tissue shape deformations. However, it has not previously been appreciated that the ECM which envelops the disc is not a passive boundary but an

active, i.e. growing material. Growth and properties of the ECM can be actively modulated by the cell layers making the ECM a controllable boundary that mechanically feeds back on tissue shape as it grows.”

10. In Figure 6 and Extended data Figure 8, the authors show that MMP2 knock-down results in a shift of the peripodial A-P boundary position. As a result, the regions chosen for the subsequent measurements in KD and control discs correspond to different medial-lateral positions within the disc. As can be seen in the authors’ own data, cell thickness differs between the medial and lateral measurement positions even within the control discs. For the data to be more convincing, the authors should compare equivalent medial-lateral positions for all measurements in control and KD discs.

- In order to appropriately compare similar regions of the PPE layer we have consistently assessed epithelial thickness 10µm posterior of the peripodial anterior/posterior boundary. This is the only landmark that we can use to fairly estimate the geometric properties of the PPE layer upon MMP2 modulation.

We think that measuring PPE thickness based on a landmark in the DP layer is not suitable since this would lead to comparing different parts of the PPE layer in control versus MMP2_{KD} discs. We have quantified PPE epithelial thickness above the DP A/P boundary and we consistently observed the same trend: While the PPE layer is thin in control discs, knockdown of MMP2 leads to a significant increase of PPE thickness (see Figure Rebuttal 3).

Figure Rebuttal 3 - Peripodial thickness above the disc proper A/P boundary

(a) Section view of control wing discs expressing RFP (green) in the posterior compartment. A/P compartment boundaries are indicated in the PPE (A/P_{PPE}) and in the DP (A/P_{DP}) by arrowheads. *right*, Magnification of the A/P boundary region, peripodial height above the A/P_{DP} is indicated by an orange arrow. (b) MMP2_{KD} wing disc, the P-compartment in which MMP2 is knocked-down is marked by RFP (green). *right*, Magnification of the A/P boundary region, peripodial height above the A/P_{DP} is indicated by an orange arrow. (c) Quantification of peripodial epithelial height above the A/P_{DP}.

11. The authors state: “In conclusion, our results indicate that growth is uniform in the plane of the DP epithelium at different stages of larval development.” This conclusion contradicts their

data in Extended Fig 3d', which shows a significant difference in proliferation rates at 95-97h AEL.

- We apologise for this imprecision. What we intended to say is that proliferation is never elevated in the centre versus the periphery of the disc. Indeed, proliferation is homogeneous at early stages, however, at later stages proliferation tends to be reduced in the centre of the disc (Supplementary Fig.3d').

We have changed the corresponding section in the main text (pg.4, line 16-18):

'Also, staining for Phospho-Histone H3, a marker for mitosis (Supplementary Fig.3d), yielded a uniform proliferation density from 72 to 80hAEL, and even a decreased central proliferation density at 96hAEL.'

Minor points:

1. The wording used to describe the changes in tissue and cell dimensions can be somewhat confusing, as the authors use both 'thickening' (at the tissue level) and 'thinning' (at the cell level) to refer to the same type of morphology change. I suggest using the words 'cell height' and 'cell width' when referring to the changes at the cell scale to clarify the authors' point.

- We appreciate this suggestion and we have changed the text using the suggested terms, now referring to changes in plane as increasing/decreasing cell width, while using thickening/thinning only for epithelial thickness changes. The section now reads (pg.3, line 29-32):

*'Consistent with the tissue scale thickening of the DP epithelium we observed that DP **cell width decreases** and **cell height increases** along the apical-basal axis. The inverse is observed for peripodial cells, which **decrease their cell height** and flatten towards the end of larval development (Supplementary Fig.S2).'*

2. The authors state: "Together, our data show a concomitant increase in tissue size with thickening and bending of the disc epithelium." This conclusion is only true for the disc proper and not for the PPE, which shows an increase in size (i.e. volume) while showing not an increase but a decrease in thickness.

- Indeed, we have not been precise here since this conclusion only holds true for the disc proper epithelium. We have corrected this now (pg.3, line33):

*'Together, our data show a concomitant increase in tissue size with thickening and bending of the disc **proper** epithelium.'*

3. The authors should be consistent in their use of the abbreviation DP. While it refers to the 'disc proper' in the main text, it is stated to refer to the 'disc pouch' in the SI on modelling. This is an important distinction when it comes to the authors' analysis of differential growth within or across epithelial layers, and should be corrected to prevent confusion.

- We thank the reviewer for this remark. By DP we mean the tissue layer between the peripodial epithelium and the bottom ECM layer, i.e. the “disc proper”. This has been corrected in the SI.

4. The authors state: “The simulations indicated that increased growth in the DP centre versus the periphery can induce tissue thickening, although a 2-fold growth mismatch generates at most a 16% increase in tissue thickness (Extended 3 Data Fig.3a-b), while experimental data indicate that the DP tissue thickness doubles.” It should be clarified why the authors chose to limit their assessment to 2-fold growth mismatch. If this number comes from experimental data, the data should be referenced. Otherwise the choice should be substantiated by argument.

- We performed these simulations to obtain intuition on the needed difference in growth magnitude between a central tissue and a peripheral tissue to obtain a thickness increase due to elastic compression. The 2-fold growth mismatch was chosen arbitrarily. However, this drastic 2-fold difference in growth magnitude only yields a marginal increase of 16% in tissue thickness.

Hence, the mentioned simulations were only meant as a qualitative, not quantitative statement. After deliberation, and in response to comments of reviewer 1, we decided that the premise of the simulations shown previously in Extended Data Fig.3a+b is not strong enough to justify its place in the article. We therefore decided to remove these simulations, which are no longer present in Supplementary Fig. 3 as well as the SI model Section M1.2. If the reviewer and editor think that these simulations are nevertheless helpful for the conceptual understanding of the reader we can include again.

5. In Extended Data Fig 4a, the authors provide images of the wing disc before and after the cut as evidence of its stressed state. However, as we can't infer the ‘nearly instantaneous relaxation’ from these snapshots, they are not very informative for making the authors' point. In addition, the provided planar view is not informative on the relaxation of the domed wing disc. If possible, the authors should provide timelapses, and section views of the discs, or should not include these images as evidence.

- We are surprised that the reviewer feels that these images do not provide useful and compelling information (see Supplementary Fig.4a). Indeed, it would be more informative to have dynamic and 3D information in tissue cuts, however this is currently not possible due to technical limitations since cutting is not compatible with confocal imaging. Independent of the static nature of the images, they clearly (for the first time) demonstrate that the wing disc accumulates residual stress during its growth. Therefore, we would like to keep this data unless the editor thinks otherwise.

6. The PPE of the collagenase treated disc shown in Figure 3a and extended Figure 5a appears to be thorn. As this may potentially have a role in the disc's relaxed state, it would be better to provide another image with the PPE intact.

- Thank you for this comment. We have replaced this image with a cross-section of an intact disc in Fig.3a and Supplementary Fig.5a.

7. In the plot in Figure 3e (as well as some others) the y axis does not start at 0. For more accurate representation of the observed changes (or lack of them), all plots of this type should start the y-axis at 0. (In several cases this would actually help accentuate the authors' conclusions).

- As suggested, we have adjusted the plots for the y-axis to start at 0.

Reviewer #3 (Remarks to the Author):

This paper addresses the 3D morphology development, such as epithelial thickening and doming, driven by elastic deformation resulting from the differential growth between the epithelial cell layer and surrounding extracellular matrix (ECM). In particular, the authors present an interesting and novel framework for how the mechanics of ECM could affect the morphology of developing organs. They demonstrate how differential growth anisotropy in the two ECM layers of *Drosophila* wing imaginal disc can lead to thickening and doming of the wing. They show that MMP2 regulates the growth anisotropy of the ECM and thus the morphology of the developing organ. To do this, the authors use experiments and a theoretical model. The experiments seem to support their claims well. The mathematical model seems relatively simple but good enough. The model appears to match the general trends from experiments but doesn't make accurate detailed predictions. Despite this, the relatively simple model does provide some novel insights into the growth induced morphogenesis. Overall, the authors provide novel insights into how ECM mechanics can affect the morphology of developing organs. Nevertheless, I'd want authors to address the following comments and answer my questions before recommending publication.

<Major>

1. What is the rationale behind PPE schematic in Fig. S2h? Volumetric increase shown in Fig. S2g doesn't back up the flattening of PPE over time.

- This figure panel *per se* is not required to understand the message of the paper. However, we have understood that it leads to confusion (also highlighted by another reviewer) and we have therefore now removed this data from the manuscript. Importantly, cell growth and cell volume are linked by division rates. In the case of the peripodial cells, it was suggested that PPE cells stop dividing and turn polyploid towards the end of disc growth. Therefore, the increased PPE cell volume could be a result of continued peripodial cell growth but a stop of cell division and polyploidy. Concerning peripodial tissue thickness we observe that peripodial flattening coincided with the doming of the DP tissue. Therefore, a likely scenario is that the PPE tissue acts mechanically as a slave to the DP/bottom ECM layers and becomes stretched due to tissue scale doming. In order to reduce the level of complexity and the amount of experimental data this we will discuss this problem in a separate manuscript.

2. In the simulation part of Fig. S3a-b, the growth rate of pouch was assumed to be higher than hinge up to twice, but according to the continued experimental results, hinge cells feature slightly better divisions and proliferation (Fig. S3c'-d'), and finally growth rate (Fig. S3g). If experimental results are the case, what's the implication of the simulation part?

- Here, we include a new panel with $\gamma_{Pouch}/\gamma_{Hinge} = 0.65$. The slowly growing pouch region turns out smaller and thinner than the fast-growing hinge region.

After deliberation, and in response to comments of reviewers 1 and 2, we decided that the premise of the above simulations is not strong enough to justify its place in the article. We therefore decided to remove these simulations, which are no longer present in Supplementary Fig. 3 as well as the SI model Section M1.2. If the reviewer and he editor think that these simulations are nevertheless helpful for the conceptual understanding of the reader we can include them again.

3. In Fig S5a, treatment of solely Collagenase reverses the curvature, whereas the combination of Y27632 and Collagenase does not. Does it mean inhibition of MyoII rather reinforces the tissue?

- Our interpretation is that full relaxation of the epithelium is only reached upon ECM digestion and concomitant MyoII inhibition. While MyoII dominates the apical cortical tension, the basal cortical tension is dominated by the ECM. In a control situation, apical and basal tension must be balanced to ensure the observed rectangular cell shape (apical and basal surface approximately similar). Inversion of tissue curvature upon ECM digestion is likely due to cell wedging caused by a strong reduction of basal tension, while apical tension dominated by MyoII remains unchanged (increased basal surface while the apical surface remains unchanged). That indeed apical MyoII contractility is responsible for curvature inversion is demonstrated by concomitant inhibition of both, the ECM and MyoII contractility. In this situation (no ECM, no MyoII), both the apical and basal cortical tension drop, likely to a similar extent such that cell shape relaxes and remains approximately rectangular, not introducing any bending.

4. In Fig. S6c and Fig. S6e, how ECM volume can be approximated as discussed in the caption? That could have caused a mismatch in Fig. S6g. Here, Fig. S1g may back up this approximation, but this comparison was done for DP, not ECM. Also, will this case still hold between integrated intensity_PPE and V_PPE?

- We have revisited ECM growth and agree that the simple approximation of ECM volume by multiplying projected/planar area with thickness is not always correct. As rightly stated by the reviewer, the approximation works well for the bottom ECM_{DP} since it is backed up by measurements of integrated Vkg::GFP intensity that lead to a similar conclusion (see Supplementary Fig.8c-d).

However, the simple approximation does not hold true for the top ECM due to *wrinkling* in the ECM layer (see Supplementary Fig.7b). In order to provide more accurate information we have now included a quantitative data set measuring growth of the top ECM_{PPE} by assessing changes in Vkg::GFP integrated intensity (as done for the ECM_{DP}). This data is now included in Supplementary Fig.8a-b. Importantly, using Vkg::GFP intensity measurement the top ECM_{PPE} shows increased growth compared to the PPE tissue layer. We find that the top ECM intensity increases ~2.5-fold more than the PPE tissue volume, likely explaining the wrinkling towards the end of development (as seen in Supplementary Fig.7b). In contrast, we find that the peripodial integrated Vkg::GFP_{PPE} intensity increases to a very similar extent as the DP volume (see Supplementary Fig.8b, magenta data points, linear fit with slope of 0.95).

This last observation indicates that the top ECM_{PPE} does not act as a geometric constraint for bending since the magnitude and the anisotropy of growth is very similar to the DP layer (and hence growth between these two layers is compatible).

This new finding is now described in the section 'Differential volume growth between DP and associated ECM' on pg.7 line 19 following:

'Next, we quantified the growth of the bottom ECM_{DP}, focusing on the wing pouch (marked by the ring of Wingless expression), following two approaches. First, we estimated the ECM_{DP} volume by multiplying the ECM thickness with the pouch area (Supplementary Fig.7d, see methods for details). Secondly, we measured the increase in integrated fluorescence intensity of the Vkg::GFP signal underlying the wing pouch (Supplementary Fig.8c). Comparing the relative increase in DP volume with the increase in ECM_{DP} estimated volume or integrated intensity showed that the DP layer outgrew the ECM_{DP} by ~15-20% (Supplementary Fig.7d and 8d). In contrast, when quantifying the changes in integrated Vkg::GFP intensity in the top ECM_{PPE} we observed that the top ECM_{PPE} intensity increases ~2.5-times faster than the PPE tissue volume (Supplementary Fig.8a-b and see discussion). Notably, we observed a very similar increase in top ECM_{PPE} intensity and DP volume (Supplementary Fig.8b), suggesting that the top ECM layer grows compatible with the DP tissue layer.

In summary, these findings show that the DP tissue layer outgrows the bottom ECM_{DP} layer, leading to an effective volumetric growth mismatch. Modelling a 20% growth mismatch was however not sufficient to account for correct tissue geometry (See Supplementary Fig.8e). In contrast, the top ECM_{PPE} layer show similar growth dynamics as the DP layer suggesting that the top ECM_{PPE} layer does not act as constraint for disc bending due to its compatible growth. This supports our hypothesis that the bottom ECM_{DP} acts as the major geometric constraint for epithelial growth.'

5. In Fig. 6e-d, it doesn't really make sense how experimental evidence after MMP2 KD, such as posterior shift of A/P_PPE, increased epithelial thickness of posterior PPE cells, and reduced apical cell surface area, can support the claim that MMP2 inhibition has to do with non-planar (3D) growth. Please elaborate on this.

- We propose the following: During wild type development peripodial cells flatten and expand to cover most of the disc proper. Upon inhibition of MMP2, because the ECM is thicker/denser this cellular and tissue expansion is inhibited: the peripodial A/P boundary is moved more posteriorly, the apical cell area is reduced and cell height increased. These three observations suggest that MMP2 interferes with the capacity of PPE cells to flatten. Given that MMP2 is a known ECM modifier, we hypothesise that reduced peripodial cell flattening in the absence of MMP2 could be due to changes in ECM growth anisotropy, i.e. the ECM could grow less in plane and constrain planar PPE cell expansion. We therefore next investigated ECM growth in the absence of MMP2 (see Fig.6). While the peripodial ECM normally grows planar (no increase in thickness), in the absence of MMP2 the peripodial ECM thickness increases, indicating non-planar 3D growth (see Fig.6f-i). We therefore conclude that MMP2 is required for planar growth of the peripodial ECM and that this allows planar expansion of the peripodial cells.

6. Why was the ECM considered an incompressible neo-Hookean material? How is this justified given the viscoelastic and fibrous nature of ECMs?

- Firstly, we measured that the ECM is incompressible. This was previously implicitly stated in the penultimate paragraph of the section “Spatial differences in ECM growth anisotropy”. For clarity, we added an explicit sentence about incompressibility there (pg.8 line 31-33):
 - *‘The area decrease and concomitant thickness increase of the bottom ECM_{DP} show a conservation of volume upon elastic deformations, which means that the ECM_{DP} is incompressible.’*
- Secondly, we use the neo-Hookean model as it is the simplest model of non-linear (finite strain) elasticity, and the typical entry point in the absence of detailed knowledge of the material. But indeed, some ECMs have been reported to have strain-stiffening properties (Han et al. 2018), which means it becomes increasingly difficult to further extend the material. Generalised neo-Hookean models that capture this effect in soft tissues are the Fung and Gent models (Goriely 2017). However, each of them introduces a strain-hardening parameter which without careful extension tests of the bottom ECM we cannot measure. Furthermore, the viscoelastic properties of active living tissues are currently a very busy area of research and capturing viscoelastic effects in a continuum model is very challenging (Erlich et al. 2022), an effort from which a well-accepted model has not yet emerged.
- We have added these points to our elaboration of model limitations in the Discussion (pg.12 line 1 following):
 - *‘We show that a nearly incompressible neo-Hookean model, in which pre-stress is added through differential growth anisotropy, can well recapitulate 3rd instar wing disc development. The assumption of incompressibility was based on measurements of ECM incompressibility in decellularization experiments (Supplementary Fig. 9c’). We use the neo-Hookean model as it is the simplest model of non-linear (finite strain) elasticity. But*

indeed, some ECMs can display strain-stiffening properties⁵³. Generalized neo-Hookean models that capture this effect (Fung and Gent models) introduce a strain-hardening parameter which requires careful extension tests of the bottom ECM that were beyond the scope of this study.

7. In Fig. 5, the model doesn't appear to match the experimental measured thickness values. Is there an explanation on why the model differs and why this doesn't affect the conclusions?

- Our model aims at providing the most simplistic solution and the minimal ingredients to explain disc thickening and bending. Indeed, we theoretically and experimentally demonstrate that two growing layers (one tissue and one ECM layer) that differ in stiffness and their magnitude and anisotropy of growth are sufficient to very quantitatively recapitulate disc morphology to a good extent. The fact that our simulations are not perfectly predicting in particular tissue thickness (Fig.5c) could have several reasons. As discussed in the previous point, for simplicity we are using a neo-Hookean model of non-linear elasticity to simulate epithelial and ECM growth. While this might be a fair choice for the cell layer, it clearly is an oversimplification concerning the complex nature of the fibrous polymer network of the ECM. Given the lack of more detailed knowledge about ECM growth, turnover-dynamics and mechanical properties we depend on taking this simplified approach here. Further simplifications are the exclusion of the surrounding hinge tissue and using simulations that are radial symmetric, neglecting that the wing pouch rather forms an oval structure. Given these simplifications it is to be expected that simulations are not perfectly recapitulating experimental data. In fact, it is satisfying to see that a rather simple approach can so well predict a complex 3-dimensional elastic growing system. Certainly, future investigations will allow us to gain a better understanding in particular of the biology and the microscopic structure of the ECM lying the wing disc and vastly improve our modelling approaches.

We have also included a section in the discussion elaborating on the modelling approach and future improvements (pg.12 line 9 following):

We aimed to keep the number of model parameters small and made several simplifying assumptions. Firstly, we decided not to model the hinge region as it grows at similar rates as the wing pouch (Supplementary Fig.3). We also assumed that the wing disc is symmetric and has a no-traction (i.e. force-free) boundary condition at the disc edge. This simplifying assumption is more correct in Anterior-Posterior direction than in Dorsal-Ventral direction, where complex folds form in larger discs⁵⁴. Capturing the whole morphology of the disc will be an interesting challenge for future studies. Nevertheless, our modelling choices, guided by relative simplicity, were sufficient to understand key physical experiments like collagenase and decellularization results and to capture quantitatively key geometric quantities and predict the degree of growth anisotropy and stiffness ratio between the layers.

<Minor>

1. In Fig. 1b, the orientation of anterior (left) is not consistent with what is explained in its caption (right).

- We appreciate that the reviewer has spotted this error. Many thanks, we have corrected it.

2. In line 21 of page 5, it says “the model predicted that the PPE layer grew ~70% more in volume than the DP layer,” but Fig. 2b notes growth rate ratio of 1.3, meaning 30%. Please clarify.

- The normalised volume ratio of PPE to DP is calculated as $\gamma_{PPE}^2/\gamma_{DP}^2 = (1.3)^2 \approx 1.70$, hence the 70%. This is explained in the SI Section M1.2.2. We slightly expanded the reasoning for calculating volume ratios that way (penultimate paragraph in M1.2.2) and added a reference to the SI in the main text at the passage quoted by the reviewer.

3. What was the choice for γ_{ECM} and γ_{DP} in Eqs. 2-3?

- This is explained in some detail in the supplementary information Section M1.2.2. Broadly, γ_{ECM} and γ_{DP} are worked out by taking the measured volumes of the bottom ECM and disc proper, while also taking into account that the growth anisotropy parameter of the ECM ρ determines how much volume is deposited in Z vs in planar direction without changing the overall ECM volume. We summarised this idea in one sentence following Eq. (3) and added a reference to the precise section in the SI.

References

- Ben Amar, Martine, and Fei Jia. 2013. "Anisotropic Growth Shapes Intestinal Tissues during Embryogenesis." *Proceedings of the National Academy of Sciences* 110 (26): 10525–30. <https://doi.org/10.1073/pnas.1217391110>.
- Daley, William P, and Kenneth M Yamada. 2013. "ECM-Modulated Cellular Dynamics as a Driving Force for Tissue Morphogenesis." *Current Opinion in Genetics & Development* 23 (4): 408–14. <https://doi.org/10.1016/j.gde.2013.05.005>.
- Erllich, Alexander, Jocelyn Étienne, Jonathan Fouchard, and Tom Wyatt. 2022. "Review: How Dynamic Prestress Governs the Shape of Living Systems, from the Subcellular to Tissue Scale." arXiv. <http://arxiv.org/abs/2209.01440>.
- Goriely, Alain. 2017. *The Mathematics and Mechanics of Biological Growth*. Interdisciplinary Applied Mathematics, volume 45. New York, NY: Springer.
- Grobstein, Clifford, and Julia Cohen. 1965. "Collagenase: Effect on the Morphogenesis of Embryonic Salivary Epithelium in Vitro." *Science* 150 (3696): 626–28. <https://doi.org/10.1126/science.150.3696.626>.
- Han, Yu Long, Pierre Ronceray, Guoqiang Xu, Andrea Malandrino, Roger D. Kamm, Martin Lenz, Chase P. Broedersz, and Ming Guo. 2018. "Cell Contraction Induces Long-Ranged Stress Stiffening in the Extracellular Matrix." *Proceedings of the National Academy of Sciences* 115 (16): 4075–80. <https://doi.org/10.1073/pnas.1722619115>.
- Hayakawa, T., J. Kishi, and Y. Nakanishi. 1992. "Salivary Gland Morphogenesis: Possible Involvement of Collagenase." *Matrix (Stuttgart, Germany). Supplement 1*: 344–51.
- Hervieux, Nathan, Mathilde Dumond, Aleksandra Sapala, Anne-Lise Routier-Kierzkowska, Daniel Kierzkowski, Adrienne H.K. Roeder, Richard S. Smith, Arezki Boudaoud, and Olivier Hamant. 2016. "A Mechanical Feedback Restricts Sepal Growth and Shape in Arabidopsis." *Current Biology* 26 (8): 1019–28. <https://doi.org/10.1016/j.cub.2016.03.004>.
- Kumar, Nilay, Kevin Tsai, Mayesha Sahir Mim, Jennifer Rangel Ambriz, Weitao Chen, Jeremiah J. Zartman, and Mark Alber. 2022. "Balancing Competing Effects of Tissue Growth and Cytoskeletal Regulation during *Drosophila* Wing Disc Development." Preprint. *Developmental Biology*. <https://doi.org/10.1101/2022.09.28.509971>.
- Lane, M.C., M.A. Koehl, F. Wilt, and R. Keller. 1993. "A Role for Regulated Secretion of Apical Extracellular Matrix during Epithelial Invagination in the Sea Urchin." *Development* 117 (3): 1049–60. <https://doi.org/10.1242/dev.117.3.1049>.
- LeGoff, Loïc, Hervé Rouault, and Thomas Lecuit. 2013. "A Global Pattern of Mechanical Stress Polarizes Cell Divisions and Cell Shape in the Growing *Drosophila* Wing Disc." *Development* 140 (19): 4051–59. <https://doi.org/10.1242/dev.090878>.
- Nematbakhsh, Ali, Megan Levis, Nilay Kumar, Weitao Chen, Jeremiah J. Zartman, and Mark Alber. 2020. "Epithelial Organ Shape Is Generated by Patterned Actomyosin Contractility and Maintained by the Extracellular Matrix." Edited by Philip K. Maini. *PLOS Computational Biology* 16 (8): e1008105. <https://doi.org/10.1371/journal.pcbi.1008105>.
- Oltean, Alina, Jie Huang, David C. Beebe, and Larry A. Taber. 2016. "Tissue Growth Constrained by Extracellular Matrix Drives Invagination during Optic Cup Morphogenesis." *Biomechanics and Modeling in Mechanobiology* 15 (6): 1405–21. <https://doi.org/10.1007/s10237-016-0771-8>.
- Sui, Liyuan, Silvanus Alt, Martin Weigert, Natalie Dye, Suzanne Eaton, Florian Jug, Eugene W. Myers, Frank Jülicher, Guillaume Salbreux, and Christian Dahmann. 2018. "Differential Lateral and Basal Tension Drive Folding of *Drosophila* Wing Discs through Two Distinct Mechanisms." *Nature Communications* 9 (1): 4620. <https://doi.org/10.1038/s41467-018-06497-3>.

REVIEWERS' COMMENTS

Reviewer #1 (Remarks to the Author):

The revision is adequate.

Reviewer #2 (Remarks to the Author):

The authors have modified the manuscript significantly and added new data that clarifies and addresses the points raised. With these changes, the authors conclusions are well supported, the paper reads much more coherent, and can be recommended for publication. However, the following points are not satisfactorily answered and should be addressed prior to publication:

1. In Figure Rebuttal 2, the authors now provide data that shows the discs indeed show growth and bending ex vivo, however for the bending, this is only provided as images from an exemplary disc. It is important that the authors provide quantification of the change in curvature, for multiple discs, comparable to those provided for discs in vivo (ex. Figure 1b).

2. In response to the question regarding the role of contractility on bending and the impact of MyoII inhibition on clone and tissue growth, the authors respond by commenting that MyoII inhibition would block cell division. Since blocking cell division does not block disc growth and development, this should not prevent the authors from performing the suggested experiments.

3. Finally, the inclusion of the rebuttal data in the manuscript is important to clarify and substantiate the authors' points. Thus, the data shown in the rebuttal figures should be incorporated into the manuscript.

Reviewer #3 (Remarks to the Author):

The author addressed all of my comments appropriately. I recommend publication.

REVIEWERS' COMMENTS

We would like to thank the referees for taking the time to re-evaluate our revised manuscript. We are glad to read that the reviewers feel that our efforts have improved the MS.

We have revisited some last questions raised by reviewer 2. Here are our point-by-point answers to the reviewer's comments:

Reviewer #1 (Remarks to the Author):

The revision is adequate.

Reviewer #2 (Remarks to the Author):

The authors have modified the manuscript significantly and added new data that clarifies and addresses the points raised. With these changes, the authors conclusions are well supported, the paper reads much more coherent, and can be recommended for publication. However, the following points are not satisfactorily answered and should be addressed prior to publication:

1. In Figure Rebuttal 2, the authors now provide data that shows the discs indeed show growth and bending *ex vivo*, however for the bending, this is only provided as images from an exemplary disc. It is important that the authors provide quantification of the change in curvature, for multiple discs, comparable to those provided for discs *in vivo* (ex. Figure 1b).

We have reanalyzed our movies and now provide quantitative information on the thickening and bending of discs in *ex vivo* culture (see Extended Data Fig.3f). We consistently observe disc bending in all observed samples. Furthermore, we refrained from directly contrasting the basal disc outlines of the *ex vivo* data with the fixed data shown in Fig.1b [despite they show bending to a very similar extent] given that this would mean comparing live tissue geometry with fixed and processed tissue.

2. In response to the question regarding the role of contractility on bending and the impact of MyoII inhibition on clone and tissue growth, the authors respond by commenting that MyoII inhibition would block cell division. Since blocking cell division does not block disc growth and development, this should not prevent the authors from performing the suggested experiments.

The reviewer is right that blocking cell division does not necessarily impair disc. Interfering with cell cycle progression via overexpression of the Retinoblastoma-family protein (RBF) leads to wing discs that harbor fewer, but bigger cells and grow to nearly normal size¹.

In striking contrast, interference with Myosin II was shown to have much stronger side effects on cell function and homeostasis than just blocking cell division. In particular, the repression of the main MyoII upstream regulators Rho1 was shown to result in strongly increased cell death (visualized by Caspase staining, see ² Fig.5A). Furthermore, interfering with Rho1 impairs several other vital processes, such as e.g. planar cell polarity³. Based on these findings it is likely that an

efficient knock-down of Rho1 should strongly impair disc growth and viability and hence rendering the proposed experiment very difficult.

3. Finally, the inclusion of the rebuttal data in the manuscript is important to clarify and substantiate the authors' points. Thus, the data shown in the rebuttal figures should be incorporated into the manuscript.

We have now included the data from the rebuttal letter into the Supplementary Figures.

In particular:

- The data on disc thickening and bending in *ex vivo* culture can be found in Supplementary Fig.3f
- The quantification of peripodial thickness above the A/P boundary is now included in Supplementary Fig.10g-i.

Reviewer #3 (Remarks to the Author):

The author addressed all of my comments appropriately. I recommend publication.

References

1. Neufeld, T. P., de la Cruz, A. F. A., Johnston, L. A. & Edgar, B. A. Coordination of Growth and Cell Division in the Drosophila Wing. *Cell* **93**, 1183–1193 (1998).
2. Neisch, A. L., Speck, O., Stronach, B. & Fehon, R. G. Rho1 regulates apoptosis via activation of the JNK signaling pathway at the plasma membrane. *J. Cell Biol.* **189**, 311–323 (2010).
3. Yan, J., Lu, Q., Fang, X. & Adler, P. N. Rho1 has multiple functions in Drosophila wing planar polarity. *Dev. Biol.* **333**, 186–199 (2009).